# 3D-MRI brain glioma intelligent segmentation based on improved 3D U-net network

**Tingting Wang** **, Tong Wu, Defu Yang, Ying Xu, Dongyang Lv, Tong Jiang, Hengjiao Wang, Qi Chen, Shengnan Xu, Ying Yan\*, Baoguang Lin\***

Department of Radiationtherapy, General Hospital of Northern Theater Command, Shenyang,China

\* yanyingdoctor@sina.com (YY), linlbg@163.com (BL)

## Abstract

### Purpose

To enhance glioma segmentation, a 3D-MRI intelligent glioma segmentation method based on deep learning is introduced. This method offers significant guidance for medical diagnosis, grading, and treatment strategy selection.

### Methods

Glioma case data were sourced from the BraTS2023 public dataset. Firstly, we preprocess the dataset, including 3D clipping, resampling, artifact elimination and normalization. Secondly, in order to enhance the perception ability of the network to different scale features, we introduce the space pyramid pool module. Then, by making the model focus on glioma details and suppressing irrelevant background information, we propose a multi-scale fusion attention mechanism; And finally, to address class imbalance and enhance learning of misclassified voxels, a combination of Dice and Focal loss functions was employed, creating a loss function, this method not only maintains the accuracy of segmentation, It also improves the recognition of challenge samples, thus improving the accuracy and generalization of the model in glioma segmentation.

### Results

Experimental findings reveal that the enhanced 3D U-Net network model stabilizes training loss at 0.1 after 150 training iterations. The refined model demonstrates superior performance with the highest DSC, Recall, and Precision values of 0.7512, 0.7064, and 0.77451, respectively. In Whole Tumor (WT) segmentation, the Dice Similarity Coefficient (DSC), Recall, and Precision scores are 0.9168, 0.9426, and 0.9375, respectively. For Core Tumor (TC) segmentation, these scores are 0.8954, 0.9014, and 0.9369, respectively. In Enhanced Tumor (ET) segmentation, the method achieves DSC, Recall, and Precision values of 0.8674, 0.9045, and 0.9011, respectively.

**Data availability statement:** All relevant data are within the paper and its Supporting information files.

**Funding:** The Liaoning Provincial Science and Technology Joint Plan (Fund);2023JH2/10170011. The funders of this study contributed to the writing and proofreading of the original paper.

## Conclusions

The DSC, Recall, and Precision indices in the WT, TC, and ET segments using this method are the highest recorded, significantly enhancing glioma segmentation. This improvement bolsters the accuracy and reliability of diagnoses, ultimately providing a scientific foundation for clinical diagnosis and treatment.

## 1 Introduction

Brain tumors, comprising abnormal cellular proliferations within the brain tissue, pose a significant hazard to human health [1]. These tumors are broadly categorized into primary and secondary types [2]. Secondary brain tumors originate from malignant neoplasms in distant body parts, such as the gastrointestinal tract, liver, or breast, which eventually infiltrate the brain. Among primary brain tumors, gliomas are the most prevalent [3]. Gliomas originate from the cancerous transformation of glial cells, which are found in the brain and spinal cord. These tumors exhibit considerable variability in size, shape, and location within the brain, making them highly individualized among patients. Accurate extraction of pertinent information pertaining to gliomas is crucial for administering appropriate treatment [4].

Magnetic resonance imaging (MRI) serves as a non-invasive modality for imaging brain tumors, enabling the detection and analysis of gliomas [5]. Its safety, risk-free nature, and provision of precise diagnostic and therapeutic insights render it an invaluable imaging technique for brain tumor diagnosis and management. Gliomas often manifest as heterogeneous tumor regions in MRI scans, typically comprising necrotic, edematous, nonenhancing, and enhancing components. Given the diverse imaging characteristics of gliomas across modalities, multimodal MRI has been employed to enhance the accuracy of glioma segmentation and the delineation of its internal structures [6].

Mishro et al. [7] employed a traditional segmentation approach, whereas deep learning has demonstrated the ability to autonomously extract more pertinent features, resulting in superior prediction outcomes. Currently, deep learning techniques have gained widespread adoption for brain tumor segmentation. Zamboglou [8] introduced a tumor segmentation method leveraging Convolutional Neural Networks (CNNs). Through the integration of a multi-channel two-dimensional (2D) CNN, the segmentation of 3D tumor regions was achieved. Nevertheless, the segmentation results were suboptimal due to the lack of optimization in the simple CNN architecture. Mathen et al. [9] pioneered the utilization of full convolutional neural networks with conditional random fields for glioma segmentation. This approach significantly enhanced segmentation accuracy, yet the complexity of the network rendered it prone to overfitting and challenges in precisely outlining tumor boundaries.

Vente et al. [10] delved into a 2D U-Net-based tumor segmentation model that incorporates densely connected blocks within the network. This design enhances information transfer and feature reuse, ultimately improving the feature extraction capabilities of the network encoder. Furthermore, the model incorporates dilation

convolution, which broadens the receptive field of the kernel without compromising resolution. To achieve fine image segmentation, a conditional random field recurrent neural network is integrated, resulting in an end-to-end trained model. However, a notable limitation is that the model converts 3D images into 2D slices, leading to a loss of spatial information and a time-consuming slice-by-slice processing approach. Billah et al. [11] introduced a 3D U-Net brain tumor segmentation model, leveraging the end-to-end deep learning network WFL. This model offers a certain degree of improvement in the accuracy of brain glioma segmentation. However, it is susceptible to interference from irrelevant background information, which contributes to a high misclassification rate and incorrect segmentation of non-tumor regions. A X Z et al. [12] proposed an efficient 3D residual neural network (ERV Net) for brain tumor segmentation. While this method shows promise, it faces challenges in effectively suppressing irrelevant background information in practical applications, thereby affecting the accuracy of segmentation results. Cao et al. [13] proposed a novel 2D-3D cascade network with multi-scale information modules. Firstly, a variational autoencoder module was integrated into the 2D-DenseUNet to regularize the shared encoder, extract useful information, and represent glioma heterogeneity; Secondly, 3D-DenseUNet and 2D network are cascaded to extract useful inter-chip features. Finally, the whole 2D-3D cascade network is trained end-to-end to make full use of 3D image information. Raza, R et al. [14] proposed an end-to-end automatic three-dimensional brain tumor segmentation (BTS) framework. The model is a hybrid of the deep residual network and the U-Net model (dResU-Net), which utilizes both low-level and high-level features to make predictions. Jia et al. [15] proposed a new end-to-end brain tumor segmentation algorithm, which embedded a coordinate attention module before sampling on the backbone network to enhance the capture ability of local texture feature information and global location feature information. Liu et al. [16] proposed a lightweight with attention mechanism of automatic 3d brain image segmentation algorithm, first of all, through the layered decoupling for standard convolution convolution, reduce the number of parameters of the model. Then, extended convolution is added to the underlying convolutional module to enhance the ability of the network to express multi-scale features. Finally, a attention mechanism is introduced in the output layer to make the network automatically focus on the tumor region.

Addressing the shortcomings of previous studies, this paper introduces an enhanced 3D U-Net network for intelligent segmentation of gliomas in 3D MRI scans. Traditional 3D U-Net networks struggle to capture contextual information of different scales simultaneously when processing brain glioma MRI data, resulting in poor segmentation performance in tumor boundaries and detail areas. This article effectively solves this problem by introducing the Atrous Spatial Pyramid Pooling Structure (ASPP). ASPP can capture multi-scale contextual information without reducing resolution through multi-scale dilated convolution operations, thereby achieving accurate segmentation of glioma regions. To further enhance the detailed information of gliomas and suppress useless background information, a multi-scale fusion attention module is introduced. This module significantly improves the sensitivity of the network to tumor details through multi-scale feature fusion and attention mechanism, while effectively reducing the interference of background noise, making the segmentation results more accurate and reliable. To tackle the issue of class imbalance and enhance the learning of misclassified voxels, we devise a loss function by combining the Dice loss and Focal loss functions. This approach not only preserves segmentation accuracy but also improves the model's ability to learn from challenging samples, leading to enhanced segmentation performance, precision, and generalization. The proposed method aims to assist physicians in making more accurate diagnostic and treatment decisions, thereby reducing misdiagnosis and missed diagnosis rates, and ultimately ensuring the authenticity and reliability of glioma diagnosis. This lays a solid scientific foundation for clinical diagnosis and treatment of gliomas. The detailed technical roadmap of our method is illustrated in Fig 1 below.

## 2 Methods

### 2.1 Dataset

The BraTS2023 public training dataset is a large glioma image dataset that contains a large number of cases, which helps the model to obtain richer sample information during training and improve its generalization ability. However, due to various factors such as device performance and patient movement, the images in this dataset may be

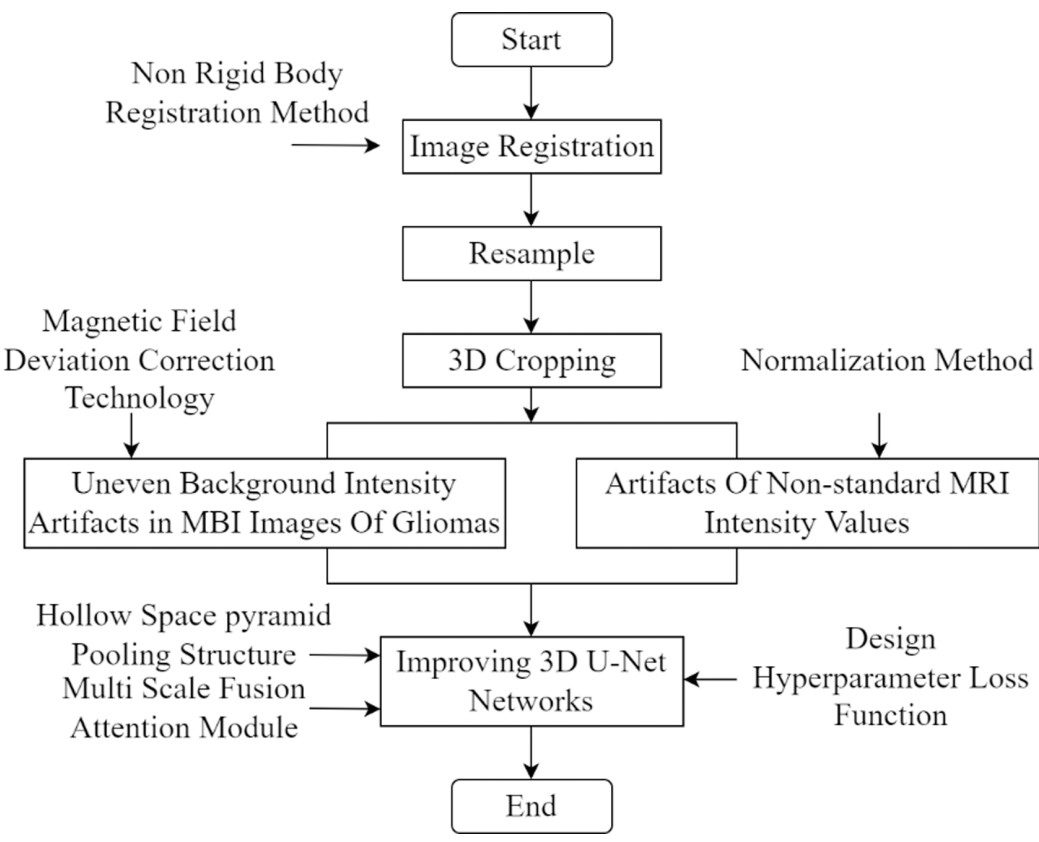

**Fig 1. Technical Roadmap.**

affected during the acquisition process, resulting in noise and artifacts in the images. Moreover, due to the large number of images in this dataset, the shape and size of its gliomas vary greatly, and the boundaries with surrounding tissues are often unclear. These factors all increase the difficulty of segmentation and affect the accurate segmentation of the model, resulting in lower accuracy and recall on the BraTS2023 dataset. Therefore, in order to further improve the performance of tumor segmentation, this study selected the BraTS2023 public training dataset to train and test the glioma segmentation model, in order to demonstrate the excellent performance of the proposed method. This dataset includes MRI data of brain tumors from over 1000 patients. It features preoperative MRI scans in four modalities: T1-weighted ($T_1$), T1 contrast-enhanced ($T_1C$), $T_2$-weighted ($T_2$), and $T_2$ fluid-attenuated inversion recovery (FLAIR). These multimodal MRI data originate from a range of clinical protocols and scanning equipment, reflecting real-world variability. All images in the dataset have undergone rigorous manual segmentation by 1-4 physicians, using a standardized labeling template, and have been validated by expert neuroradiologists. The labeled regions encompass the enhancing tumor (ET, labeled 4 in red), peritumoral edema (ED, labeled 2 in green), necrotic and non-enhancing tumor core (NET, labeled 1 in blue), and non-tumor areas (NT, labeled 0). These labels serve as the gold standard (Ground Truth, GT) for training and evaluating our model. To mitigate local image deformation, we employed non-rigid registration methods to align the four preoperative MR modalities with standard space reference images. During the registration process, the deformation parameters of the input image were continuously adjusted to minimize the difference between them and the reference image. The specific implementation process is described as follows:

During the registration process, the mean square error (MSE) is used to measure the similarity between the input image and the reference image, with the goal of minimizing this MSE value. The mean square error is expressed as follows:

$$MSE = \frac{1}{N} \sum_{i=1}^{N} (I_1(i) - I_2(i))^2$$

(1)

In the formula, $I_1(i)$ is the input image; $I_2(i)$ is the reference image; $N$ is the total number of pixels.

Registration continuously adjusts the deformation parameters of the input image to minimize the difference between it and the reference image. This deformation parameter is usually represented by a displacement vector field, namely deformation field $d(x,y,z)$, which describes the deformation mapping from the input image to the reference image, that is, each pixel (or voxel) in each direction has a corresponding displacement vector. $d(x,y,z)$ represents as follows:

$$d(x,y,z) = mathbf(d_x(x,y,z), d_y(x,y,z), d_z(x,y,z))$$

(2)

To prevent excessive deformation, a regularization term is added during the registration process to constrain the smoothness of the deformation field. The regularization term is represented as follows:

$$E_{reg} = \int \|\nabla d(x,y,z)\|^2 dxdydz$$

(3)

In the equation, $\nabla d(x,y,z)$ is the gradient of the displacement vector; $\|\nabla d(x,y,z)\|$ is the modulus of the gradient vector. Then, through iterative optimization, the deformation parameters are continuously adjusted. In each iteration, the similarity measure and regularization term will be calculated based on the current deformation parameters, and the deformation parameters will be updated using gradient descent until the MSE value of the similarity measure reaches the minimum and satisfies the constraint of formula (3), ending the iteration and completing the registration.

Subsequently, based on the registered images, skull stripping was performed to isolate the brain region, and the brain images were resampled to a voxel resolution of 1mm × 1mm × 1mm, ensuring consistency across the dataset.

## 2.2 Data pre-processing

To fully exploit the rich 3D spatial information contained in the multimodal glioma MRI data, we tailored the MRI images and corresponding masks based on the precise boundary contours of the 3D brain tissue. Specifically, we resampled these images to a standardized resolution of 128 × 128 × 128 voxels, ensuring consistency and optimal use of the volumetric data. MRI scans are often subject to artifacts due to patient motion and magnetic field inhomogeneities. These artifacts primarily manifest as background intensity inhomogeneities and non-standard MRI intensity values, which can significantly impact the accuracy of subsequent analysis. To mitigate these artifacts, we implemented an N4ITK-based magnetic field bias correction technique. This approach effectively corrected the background intensity inhomogeneity artifacts present in the glioma MRI images, enhancing image quality. Additionally, to address non-standard MRI intensity artifacts, we employed a normalization method, further refining the image data for more reliable analysis.

$$x_{norm} = \frac{x - \mu}{\sigma}$$

(4)

Where the MRI intensity values for each glioma pattern are denoted as $x$; the mean and standard deviation of the intensity values for the entire brain region in the MRI images of gliomas are represented by $\mu, \sigma$; and the normalized intensity values are truncated in the range of [-5, 5], and finally normalized to the interval of [0,1].

Based on the above corrections, in order to further avoid the impact of noise on the subsequent segmentation performance of MRI images, we now proceed with processing according to the theory of stochastic resonance. Random resonance is a nonlinear phenomenon, and under certain conditions, the addition of noise can enhance the detection ability of weak signals. In MRI images, although noise is often considered interference, it is possible to transform noise into a factor that helps improve image quality by utilizing stochastic resonance theory. From the perspective of signal processing, the signal in MRI images can be seen as a mixed signal consisting of target tissue information (useful signal) and noise and artifacts (interference signal). The theory of stochastic resonance can enhance useful signals in the presence of noise by adjusting certain parameters of the system. The specific implementation process is as follows:

Firstly, a method based on local statistical information is used to estimate the noise level in MRI images, as shown below:

$$D = k\sigma^2 \tag{5}$$

In the formula, $D$ represents the noise intensity; $k$ is the adjustment coefficient, which can take values between 0.1-10; $\sigma^2$ is the noise variance.

Secondly, the modified and normalized data mentioned above is used as the amplitude $A$ of the input MRI image signal, which is then fed into a bistable system. The dynamic equation can be expressed as:

$$\frac{dx}{dt} = x - x^3 + A\sin(\omega t) + \sqrt{2D}\xi(t) \tag{6}$$

In the formula, $x$ is the state variable of the system; $\omega$ is the angular frequency of the input signal, with an initial value of 1, and then adjusted according to the processing result; $\xi(t)$ is Gaussian white noise.

Then, numerical methods are used to solve the dynamic equations of the bistable system, and the output signal after stochastic resonance processing is obtained. For the grayscale value of each pixel, the value of the system's state variable $x$ is gradually calculated according to a set time step (such as $\Delta t = 0.01$). After a certain period of time (such as $T = 100$ time units), the stable value of $x$ is taken as the processed grayscale value of the pixel, and the corresponding output signal is obtained.

Finally, the one-dimensional signal processed by stochastic resonance is converted back into a two-dimensional image. Evaluate the processed image using the Peak Signal to Noise Ratio (PSNR) image quality assessment metric. If the evaluation indicators do not improve, the parameters of the stochastic resonance system (such as noise intensity $D$, angular frequency $\omega$) can be adjusted and reprocessed until satisfactory image quality improvement is achieved.

## 3  3D-MRI brain glioma intelligent segmentation network model based on improved 3D U-Net

To improve segmentation performance, this paper enhances the 3D U-Net network by integrating Atrous Spatial Pyramid Pooling (ASPP) and a multi-scale fusion attention module (MFAB). The ASPP module captures contextual information across multiple scales in 3D images, while the MFAB module focuses on identifying salient features of brain glioma regions in 3D-MRI scans and suppressing irrelevant areas. Furthermore, a combined loss function, which incorporates both Dice and Focal loss functions, is introduced to further enhance the segmentation accuracy of 3D-MRI brain glioma.

### 3.1  3D U-Net network

The 3D U-Net network, a stereoscopic segmentation network designed for learning from sparsely labeled stereo images [17,18]. The 3D U-Net network excels in learning from limited data, and its multi-layer convolution structure efficiently reduces data volume without compromising key features, thereby enhancing learning efficiency [19,20]. The 3D U-Net model comprises two segments: the encoding and decoding parts [21,22].

## 3.2 Pyramid pooling of empty space

Building on the established 3D U-Net network structure, this study introduces the ASPP module to enhance the network and improve the accuracy of brain glioma segmentation. The ASPP module can capture contextual information at different scales in 3D images and help the network better capture different parts of the tumor, thereby improving the quality of brain glioma segmentation. While atrous convolution expands the receptive field of the convolution kernel, increasing network layers may lead to loss of image information and the risk of overfitting. Utilizing a receptive field of uniform size limits the extraction to single-scale features [23], as depicted in Fig 2. The ASPP structure, known for its robustness in capturing multi-scale context information, excels in multi-scale object segmentation [24]. ASPP combines atrous convolution with spatial pyramid pooling, featuring multiple parallel atrous convolutions at varying dilation rates. Smaller dilation rates correspond to pixels near the image's center, while larger rates pertain to more distant pixels. By concatenating feature maps from atrous convolutions with different dilation rates, multi-scale information encoding is achieved. This allows output feature map neurons to contain multiple receptive fields of varying sizes, thereby improving segmentation performance.

The ASPP module's architecture includes a 1×1×1 convolution, three parallel 3×3×3 atrous convolutions with dilation rates of 6, 12, and 18, and a global average pooling layer. Cascading these components and passing them through a 1×1×1 convolution reduces the number of channels to the desired value, effectively capturing the contextual information of 3D-MRI glioma images at different scales.

## 3.3 Multi scale fusion attention module

The attention mechanism in deep learning, analogous to the human visual system, acts as the visual component of a deep learning mode l [25]. In brain glioma segmentation, challenges arise from the characteristics of gliomas in 3D-MRI

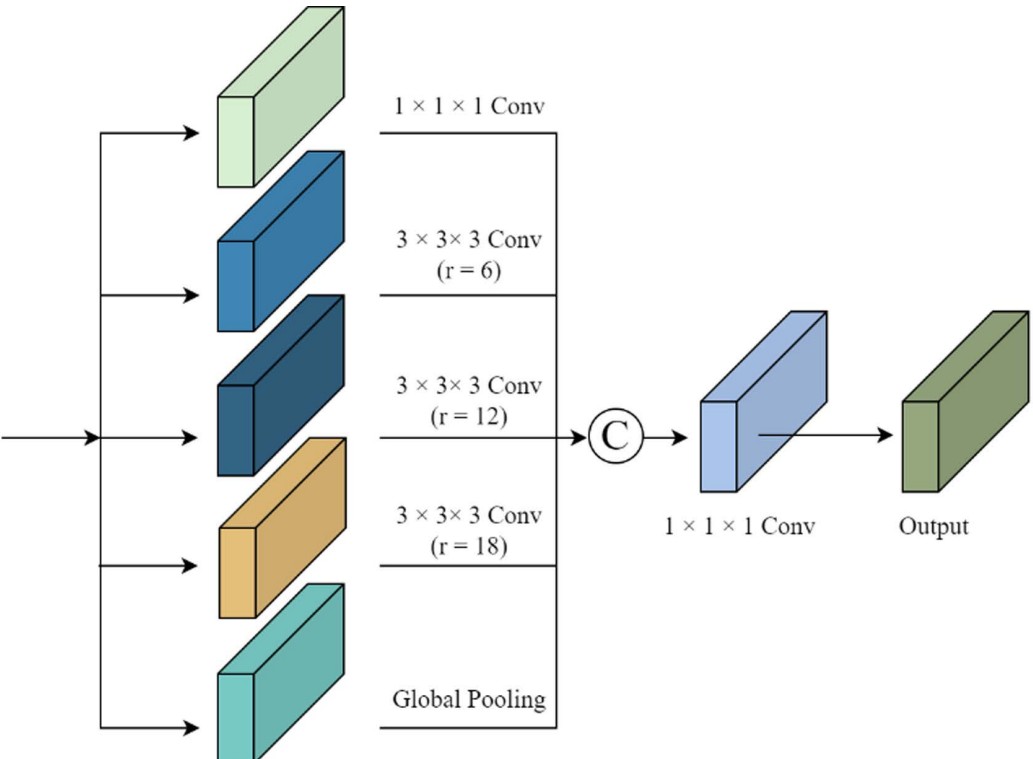

**Fig 2. ASPP Basic Structure.**

images: gliomas, typically smaller than the entire image, contribute to class imbalance and reduce segmentation accuracy. Their variable shapes and indistinct boundaries often lead to over-segmentation issues [26]. To enhance segmentation, this paper integrates a MFAB into the 3D U-Net network. This module accentuates glioma details, suppresses irrelevant background information, and improves the model's segmentation accuracy and generalization capability. The fundamental structure of this module is illustrated in Fig 3. The MFAB module utilizes a channel attention mechanism to capture inter-dependencies between feature map channels, thereby enhancing the representation of brain glioma features in 3D-MRI images [27]. Additionally, research indicates that leveraging multi-scale image information can enhance model segmentation accuracy. Consequently, the MFAB block incorporates multi-scale data to extract glioma features in 3D-MRI.

The MFAB module discerns dependencies between image feature channels by amalgamating high-dimensional and low-dimensional feature maps. Mirroring the human visual system, it automatically selects information crucial for glioma segmentation. The core concept behind the MFAB design involves learning the significance of each feature map channel at different levels and amplifying useful feature maps without adding extra spatial dimensions. Concurrently, feature maps contributing less to glioma segmentation can be suppressed based on their relevance. In this research, the interdependence of feature map channels is established using both low and high-dimensional feature maps. High-dimensional features contain rich semantic information, while low-dimensional features, often overlooked, provide extensive edge details, aiding in image detail restoration. The basic structure of the MFAB model is depicted in Fig 3.

The goal of processing high-dimensional and low-dimensional features via a channel attention mechanism is to amplify the significance of each feature channel's critical information and eliminate redundant features during 3D-MRI glioma segmentation. Initially, the high-dimensional feature map $XH^*_{input}$ passes through 1×1 and 3×3 convolutional layers, yielding a characteristic pattern $XH_{input}$, which can be described as:

$$F_{Cov} : XH^*_{input} \rightarrow XH_{input}$$

(7)

Among them, $XH^*_{input} \in R^{H \times W \times C}$, $XH_{input} \in R^{H \times W \times C}$. The characteristic diagram $XH_{input}$ and low-dimensional feature maps $XL_{input}$ has the same number of channels. $V = [v_1, v_2, \cdots, v_c]$ is defined as a set containing convolution layers, $v_c$ represents $c$ the number of parameters for each roll up layer. Output set $U = [u_1, u_2, \cdots, u_c]$ the mathematical formula of is described as:

$$u_c = v_c * X_{input} = \sum_{i=1}^{C} \left( v_c^j \right) * x^i$$

(8)

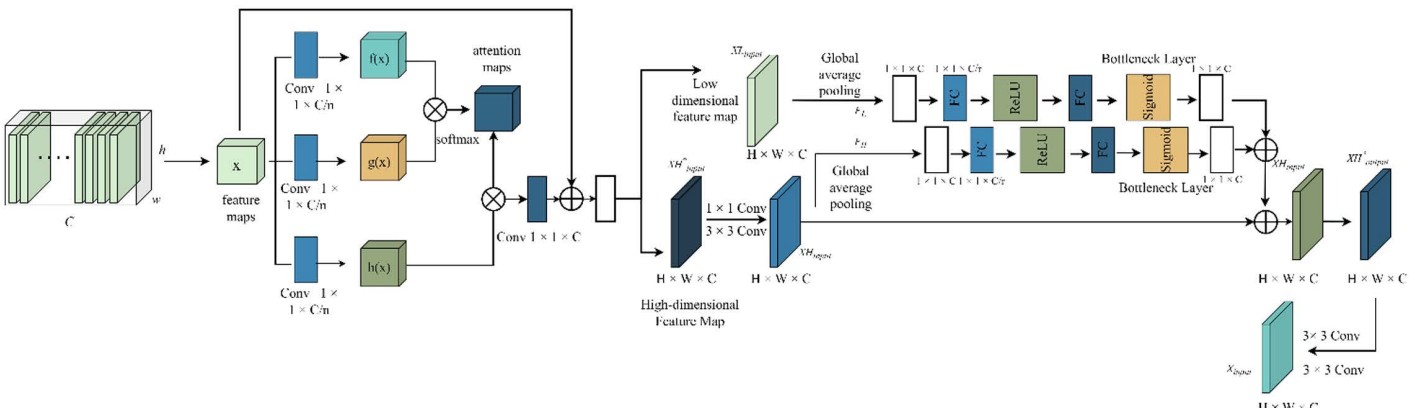

**Fig 3. Basic structure of MFAB module.**

Including: $v_c = [v_c^1, v_c^2, \cdots v_c^c]$, $X_{input} = [x^1, x^2, \cdots x^c]$, $X_{input} \in (XH_{input} \text{ or } XL_{input})$; $*$ represents a convolution operation.

Channel aggregation is performed using a global average pooling layer to obtain channel-based statistics 1×1×C. $XH_{input}$ and $XL_{input}$ post-processing with the global average pooling layer, statistics can be obtained $S_{c1}$ and $S_{c2}$. The formulas are described as follows:

$$S_{c1} = F_L(XL_{input}) = \frac{\sum_{i=1}^{H} \sum_{j=1}^{w} u_c(i,j)}{H \times W} \tag{9}$$

$$S_{c2} = F_H(XH_{input}) = \frac{\sum_{i=1}^{H} \sum_{j=1}^{w} u_c(i,j)}{H \times W} \tag{10}$$

Including: $F_L$ denotes the global average pooling operation for low-dimensional feature maps, $F_H$ represents the global average pooling operation for high-dimensional feature maps; The feature map's height and width are denoted as $H$ and $W$; For characteristic diagram of each channel $u_c$ express; $(i,j)$ pixels representing the feature map; Then, through the bottleneck layer with full connection layer and activation function, the complexity of 3D U-Net glioma segmentation model is improved by using the bottleneck layer constraints, while capturing low and high-dimensional channel feature maps $z_1$ and $z_2$. The computational formulas for low-dimensional and high-dimensional channel feature maps are as follows:

$$z_1 = F_{Ls}(S_1, P) = \delta_1(P_1 \delta_2(P_2, S_1)) \tag{11}$$

$$z_2 = F_{Hs}(S_2, P) = \delta_2(P_1 \delta_2(P_1, S_2)) \tag{12}$$

Among them, the bottleneck layer manipulates high and low-dimensional feature maps using $F_{Hs}$ and $F_{Ls}$ to express; The full connected layer is denoted as $P_1$ and $P_2$; For Sigmaid function use $\delta_1$ to express; ReLu activation function use $\delta_2$ to express.

Utilize $F_{add}$ function is employed to merge high and low-dimensional feature maps, and the formula is described as:

$$z = F_{add}(z_1 + z_2) \tag{13}$$

With activation function $\beta$ scale transformation of $T$ determine $XH^*_{output}$, the formula is described as:

$$XH^*_{output} = F_{scale}(T_c, \beta_c) = \beta_c T_c \tag{14}$$

Including: $XH_{output-c} = [\tilde{XH}_{output-1}, \tilde{XH}_{output-2}, \cdots, \tilde{XH}_{output-c}]$; $\beta_c$ is the scale transformation factor, $T_c \in R^{H \times W}$ is a characteristic graph and can be determined by matrix multiplication $F_{scale}(T_c, \beta_c)$. This multiplication facilitates the recalibration of original channel dimension features. To bolster feature representation and enrich semantic information, it can be combined through channel superposition $XH_{input}$ and $XH_{output}$ realize the determination of $XH^*_{output}$. Multi-scale fusion attention module output $XH^*_{output}$. Through two 3×3 convolutional layers are obtained, for the purpose to further capturing semantic information.

## 3.4 Improved 3D U-Net network segmentation model

Building on the previous discussions, the enhancement of the 3D U-Net network is achieved, with the basic structure of the improved 3D U-Net network segmentation model depicted in Fig 4. This model comprises a contraction path for extracting features from the input image and an expansion path for decoding these features and integrating them with

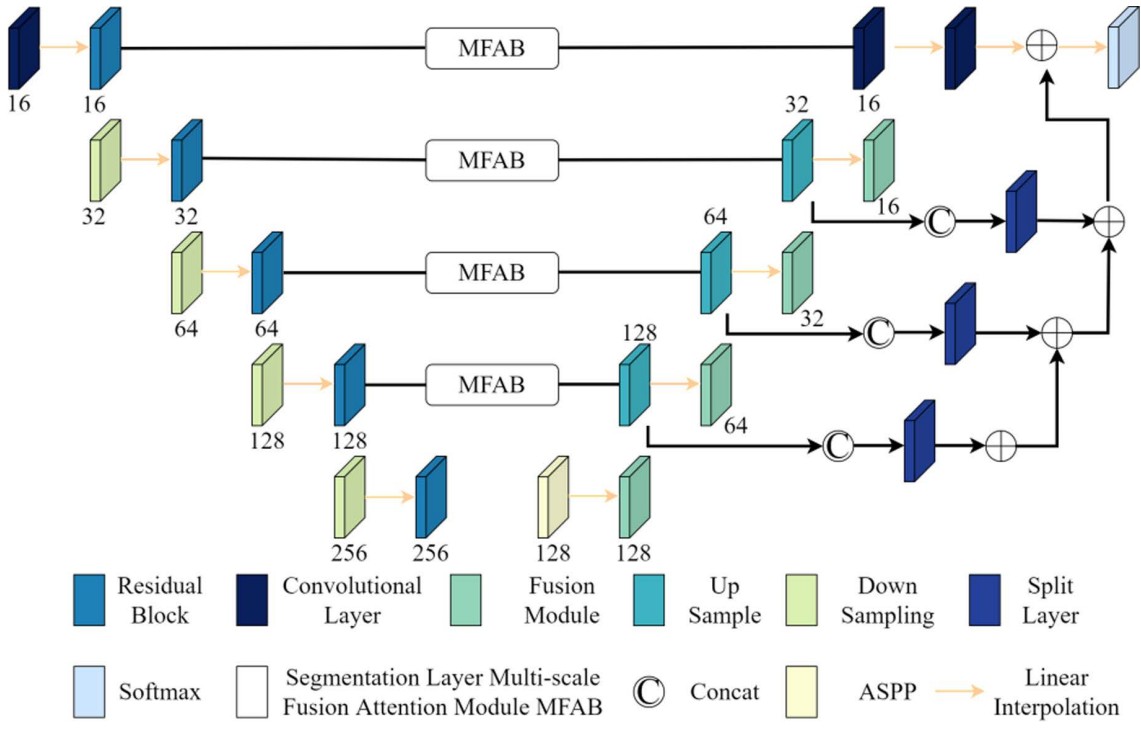

**Fig 4. Improve the basic structure of 3D U-Net network.**

low-level features to precisely locate the target structure. The network processes four modal 3D-MRI images, each of size 128×128×128. The contraction path includes multiple 3×3×3 residual blocks. These are linked to the dropout layer via a down-sampling module, reducing the feature map resolution and accommodating more features along the aggregation path. Thus, the refined 3D U-Net network model facilitates intelligent segmentation of 3D-MRI glioma. The specific implementation is as follows: ASPP is applied in the contraction path to generate a feature map, which is then input into the expansion path to capture multi-scale features, enhancing accuracy. The spatial resolution in the contraction path is reduced fourfold due to 2-stride convolutions, resulting in a spatial resolution 16 times smaller than the input. The expansion path utilizes up-sampling modules and fusion modules to reverse the contraction path process and restore the feature map resolution. At each layer, the contraction path output is combined with the up-sampling layer output from the expansion path. Before this combination, an attention mechanism is used to suppress irrelevant glioma tissue information in the contraction path output, ensuring that feature vectors at various resolutions positively influence the final output. The expansion path's up-sampling module comprises 3×3×3 and 1×1×1 convolution layers. Post-combination, the fusion module reconfigures features, further reducing the number of feature maps, crucial for minimizing memory usage. Different layers of the expansion path use a 1×1×1 convolution kernel, generating multiple segmentation maps at varying resolutions. These maps are adjusted to corresponding resolutions through linear interpolation and combined via element-wise summation to produce the final network output. Compared to the more common transposed convolution, this approach offers similar performance while preventing the network output from developing checkerboard artifacts.

Within the entire network, the Leaky ReLu activation function is utilized to process all convolution characteristic graphs. Moreover, traditional batch normalization has been substituted with Instance Normalization to offset the randomness introduced by small batch training, a necessity due to memory limitations. And to cope with the large amount of memory and processing power required for computationally intensive models, the concept of parallel computing is used to overcome

this problem. The atrous spatial pyramid pooling structure introduced in the 3D U-Net network and the network structure of the multi-scale fusion attention module are run separately on different GPUs, and data transmission is carried out through high-speed communication links between the atrous spatial pyramid pooling structure and the network structure of the multi-scale fusion attention module to ensure communication and data synchronization between different GPUs. And the tasks in the multi-scale fusion attention module can also be decomposed into smaller tasks, that is, the calculation of different scale features can be performed in parallel on different computing units, and then the fusion operation can be performed. Based on this, to some extent, the problem of high memory and processing power requirements brought by the model can be overcome, and overall efficiency can be improved.

### 3.5 Loss function

Given that image segmentation is conducted on a pixel-by-pixel basis, with larger regions exerting a more significant influence on the loss [28], a loss function is employed to enhance the segmentation accuracy of 3D-MRI for glioma. This function comprises both the Dice loss and Focal loss functions. Train the improved 3D U-Net network model with the loss function as the optimization objective to comprehensively consider the accuracy of prediction results and class imbalance issues, thereby improving the model's generalization ability and ensuring the accuracy of 3D MRI segmentation of gliomas. The Dice loss function, which is the value obtained by subtracting the Dice coefficient from 1, transforms the voxel level labeling problem into a problem of minimizing inter class distance to learn the distribution of classes, and to some extent solves the problem of class imbalance; The Focal loss function changes the sample weights to make the model more inclined to focus on difficult to classify samples during training, in order to better learn voxels with classification errors. Formulas (15)–(17) define paragraphs $b$ the probability of false positive, false negative, and false positive:

$$TP_p(b) = \sum_{n=1}^{N} p_n(b) g_n(b) \tag{15}$$

$$FN_p(b) = \sum_{n=1}^{N} (1 - p_n(b)) g_n(b) \tag{16}$$

$$FP_p(b) = \sum_{n=1}^{N} p_n(b) (1 - g_n(b)) \tag{17}$$

Where: The probability of voxel $n$ of $b$ is denoted as $p_n(b)$; The probability of voxel annotated by experts $n$ of $b$ is expressed as $g_n(b)$; The number of $b$ that the probability that the class is true positive is expressed as $TP_p(b)$; Section $b$ that the probability that the class is true negative is expressed as $FN_p(b)$; Section $b$ that the probability that the class is false positive is $FP_p(b)$; The voxel number of 3D-MRI image blocks for glioma use $N$ to express.

The definition formula of the loss function is described as:

$$L = L_{Dice} + L_{Focal} = B - \sum_{b=0}^{B-1} \frac{TP_p(b)}{TP_p(b) + \alpha FN_p(b) + \beta FP_p(b)} - \frac{\lambda \cdot \sum_{b=0}^{B-1} \sum_{n=1}^{N} g_n(b)(1 - p_n(b))^2 \lg(p_n(b))}{N} \tag{18}$$

Wherein: the weights to achieve the balance between false negative and false positive are respectively used $\alpha$ and $\beta$ to express; The number of categories including background is expressed as $B$; The weight value to realize the balance of Dice loss and Focal loss is $\lambda$.

# 4 Results and discussion

We use Dice Similarity Coefficient (DSC), Hausdorff distance (HD95), recall, and precision for evaluation. The specific formula for calculating DSC is:

$$D = \frac{2\left|G \cap O\right|}{\left|G\right| + \left|O\right|} \tag{19}$$

Including: $G$ represents the actual value label, $O$ is the segmentation result of the evaluation method. The value range of the DSC index is (0,1). A value of 0 indicates no overlap between the segmentation result and the actual clinician-marked area, whereas a value of 1 signifies complete overlap. Clearly, the higher the DSC value, the more accurate the brain tumor segmentation result.

$$Recall = \frac{\left|G \cap O\right|}{\left|G\right|} \tag{20}$$

$$Precision = \frac{\left|G \cap O\right|}{\left|O\right|} \tag{21}$$

Using the BraTS2023 public training set as the research object, 60 glioma samples were randomly selected as the test set, including 15 cases of LGG and 45 cases of HGG. 80% of the remaining samples were used as the training set and 20% as the validation set. Among them, the number of samples in the training set is 180, including 55 cases of LGG and 125 cases of HGG; The sample size in the validation set is 45 cases, including 18 cases of LGG and 27 cases of HGG. To solve the problem of overfitting and improve the generalization ability and robustness of the model, the BraTS2023 public training set was used as the basis, and the BraTS2021 dataset was introduced for training, which also contains a large number of annotated brain glioma MRI images. Thus, 300 samples from BraTS2021 were introduced, and the two training sets were merged to expand to 480 samples. The intelligent segmentation experiment of 3D-MRI glioma was conducted using the Pytorch framework. The Adam optimizer was chosen for network weight updates, with a learning rate set at 0.001 and a weight decay of 10-4. L2 regularization was implemented. To minimize network computation and limit performance impact, MRI images were cropped to a size of 128×128×128, aiming to exclude as much background area as possible. The proposed method was applied to 3D-MRI glioma segmentation, and its performance was analyzed.

The efficacy of the brain glioma segmentation model significantly influences its segmentation accuracy. Utilizing the 3D U-Net network as a foundational structure, this paper introduces the ASPP network and a multi-scale fusion attention module, MFAB, to conduct ablation studies. Three brain glioma segmentation network models were developed: the 3D U-Net network model, the 3D U-Net+ASPP network model, and the enhanced 3D U-Net network model. These models were trained using 180 samples from the BraTS2023 dataset(BraTS 2023 Challenge - syn51156910 - Wiki) [29–31], and by comparing the loss changes of each network model in the validation set and the test set, the performance of each network model was studied. The experimental results are shown in Table 1.

According to Table 1, there is a significant difference in the performance of the 3D U-Net network model and the 3D U-Net+ASPP network model between the validation set and the test set, indicating the presence of overfitting. Among them, in the test set, the 3D U-Net network model and the 3D U-Net+ASPP network model showed a decreasing trend with increasing training times. Compared to the losses of the three models, the improved 3D U-Net network model has lower losses than the 3D U-Net network model and the 3D U-Net+ASPP network model. In the test set, as the number of iterations increases, the loss of each glioma segmentation network model shows a decreasing trend and ultimately

**Table 1. Change analysis of loss function before and after improvement of glioma segmentation model.**

| Dataset | Epochs | Losses | | |
|---|---|---|---|---|
| | | 3D U-Net model | 3D U-Ne+ASPP model | Improved 3D U-Net Model |
| Validation set | 0 | 0.77 | 0.58 | 0.50 |
| | 80 | 0.58 | 0.39 | 0.18 |
| | 160 | 0.50 | 0.35 | 0.10 |
| | 240 | 0.54 | 0.40 | 0.10 |
| | 320 | 0.59 | 0.43 | 0.10 |
| Test set | 0 | 0.78 | 0.65 | 0.52 |
| | 80 | 0.59 | 0.41 | 0.19 |
| | 160 | 0.51 | 0.41 | 0.11 |
| | 240 | 0.50 | 0.36 | 0.11 |
| | 320 | 0.50 | 0.36 | 0.11 |

remains stable. The improved 3D U-Net network model showed a rapid decrease in loss during the first 80 iterations, with a significant decrease. After increasing the number of iterations, the model loss exhibited a lower degree of change. When the number of iterations reached 160, the model loss began to stabilize, with a final loss of 0.11, which was not significantly different from the representation in the validation set; During the iteration process, the loss of the 3D U-Net network model significantly decreased in the first 80 iterations. Before reaching 260 iterations, the loss value showed a slow downward trend, and the model slowly reached a convergence state after 260 iterations. Its loss was much higher than that of the improved glioma segmentation model, with a loss of 0.50, but there was a significant gap compared to its performance in the validation set. The loss of the 3D U-Net+ASPP network model is lower than that of the original 3D U-Net network model due to the introduction of ASPP structure. After 240 iterations, it stabilizes at 0.36, which is significantly different from the performance page in the validation set. The experimental results show that, based on the 3D U-Net network model, the introduction of ASPP structure and multi-scale fusion attention module MFAB can effectively learn the features of glioma MRI images from training samples, eliminate the influence of irrelevant information on segmentation results, and introduce additional datasets during training to improve the generalization ability and robustness of the model, thereby enhancing the glioma segmentation effect. This indicates that the improved 3D U-Net network model has outstanding performance.

To further validate the performance of the improved 3D U-Net network model, different levels of noise were introduced based on the test set to test its robustness in the face of interference. The test results are shown in Table 2.

According to the analysis of Table 2, it can be seen that the losses of the three models all show an upward trend when facing different levels of noise interference. The improved 3D U-Net network model can maintain a loss below 0.15 under 5-25dB noise interference, and the increase in loss is relatively stable. When the noise interference exceeds 25dB, there is a significant increase in loss. When it reaches 35dB, the loss gradually stabilizes and the increase decreases. When the noise interference exceeds 45dB, the loss increases rapidly, but it still remains below 0.35. However, the 3D U-Net network model shows a significant increase in loss when facing noise interference, and does not have sufficient ability to adapt to noise interference. The 3D U-Net+ASPP network model shows a gradual increase in loss under 5-10dB noise interference, which can be maintained below 0.8. However, when the noise interference exceeds 10dB, the loss shows a rapid increase trend, and after reaching 15dB, the loss shows an upward gradual increase. The 3D U-Net+ASPP network model is capable of extracting sufficient information from data for accurate segmentation while maintaining low loss when noise interference is low. Due to its segmentation based on image features, when the noise interference exceeds a certain value, the model may not be able to extract sufficient useful features corresponding to the target, resulting in increased loss. The results show that introducing ASPP structure and multi-scale fusion attention module MFAB on the basis of 3D

**Table 2. Analysis of Changes in Loss Function under Different Levels of Interference before and after Improvement.**

| Noise interference/dB | Losses | | |
|---|---|---|---|
| | 3D U-Net model | 3D U-Net+ASPP model | Improved 3D U-Net Model |
| 5 | 0.90 | 0.69 | 0.04 |
| 10 | 1.16 | 0.79 | 0.06 |
| 15 | 1.24 | 1.09 | 0.08 |
| 20 | 1.42 | 1.19 | 0.10 |
| 25 | 1.54 | 1.25 | 0.12 |
| 30 | 1.61 | 1.35 | 0.16 |
| 35 | 1.70 | 1.42 | 0.19 |
| 40 | 1.83 | 1.51 | 0.22 |
| 45 | 1.95 | 1.59 | 0.27 |
| 50 | 2.11 | 1.62 | 0.32 |

U-Net network model helps the model to have better perception and expression ability in dealing with complex scenes and low noise interference, and can better focus on the target area, reduce sensitivity to noise, and improve robustness, so that it can maintain low loss under a certain range of noise interference. The improved 3D U-Net network has outstanding performance, strong robustness and generalization ability, and can effectively improve the segmentation effect of glioma.

Based on the above, to further demonstrate the robustness of the proposed method, 5 samples of LGG and 5 samples of HGG were randomly selected from the training set, validation set, and testing set. Using the Jaccard Intersection over Union (IoU) index, the results of the 3D U-Net model, 3D U-Net+ASPP model, and improved 3D U-Net model on the training set, validation set, and testing set were statistically analyzed to evaluate the processing ability of each method on samples from different datasets. Among them, the IoU index is similar to the Dice coefficient, which also measures the overlap between the segmented area and the real area. The value range of IoU is between 0 and 1, and the larger the value, the higher the degree of overlap and the more accurate the segmentation result. The specific results are shown in Table 3.

According to the results in Table 3, it can be seen that in the training set, whether it is LGG or HGG type gliomas, the IoU values of the 3D U-Net model, 3D U-Net+ASPP model, and improved 3D U-Net model are higher than those in the validation set and test set, with LGG having a higher IoU value than HGG. Comparing the IoU values of different types of gliomas in the 3D U-Net model, 3D U-Net+ASPP model, and improved 3D U-Net model, the IoU value of the improved 3D U-Net model is higher than that of the 3D U-Net model and 3D U-Net+ASPP model. Moreover, the IoU value changes slightly in different datasets, with a maximum difference of 0.03 in the results. The 3D U-Net model and 3D U-Net+ASPP model have higher IoU values in the training set, but once applied to the validation set and test set, their IoU values undergo significant changes. Based on the above analysis, it can be concluded that the improved 3D U-Net model has strong processing ability for samples from different datasets, high segmentation accuracy, and good robustness.

Utilizing DSC, Recall, and Precision as evaluation metrics, the 3D U-Net network model, the 3D U-Net+ASPP network model, and the enhanced 3D U-Net network model were employed to segment the 3D-MRI brain glioma of training samples. An analysis of the changes in DSC, Recall, and Precision values for each network model validated the efficacy of the enhanced 3D U-Net network model in brain glioma segmentation. The results are presented in Table 4.

Table 4 analysis reveals that when segmenting 3D-MRI glioma, the DSC, Recall, and Precision metrics of the enhanced 3D U-Net network model surpass those of both the 3D U-Net and 3D U-Net+ASPP network models, with the 3D U-Net model scoring the lowest. Higher index values indicate superior network model performance, further corroborating that the enhanced 3D U-Net network model significantly improves glioma segmentation and possesses practical applicability.

**Table 3. Results of Jaccard Index Index under Different DataSets.**

| Dataset type | Types of glioma | Sample number | IoU value | | |
| --- | --- | --- | --- | --- | --- |
| | | | 3D U-Net model | 3D U-Ne+ASPP model | Improved 3D U-Net Model |
| Training set | LGG | 1 | 0.96 | 0.97 | 1.00 |
| | | 2 | 0.97 | 0.98 | 1.00 |
| | | 3 | 0.96 | 0.97 | 1.00 |
| | | 4 | 0.96 | 0.97 | 0.99 |
| | | 5 | 0.97 | 0.98 | 1.00 |
| | HGG | 1 | 0.95 | 0.96 | 0.98 |
| | | 2 | 0.96 | 0.97 | 0.99 |
| | | 3 | 0.95 | 0.96 | 1.00 |
| | | 4 | 0.96 | 0.97 | 0.99 |
| | | 5 | 0.96 | 0.97 | 1.00 |
| Validation set | LGG | 1 | 0.92 | 0.93 | 0.98 |
| | | 2 | 0.94 | 0.95 | 0.99 |
| | | 3 | 0.90 | 0.91 | 0.98 |
| | | 4 | 0.89 | 0.90 | 0.98 |
| | | 5 | 0.91 | 0.92 | 0.99 |
| | HGG | 1 | 0.87 | 0.90 | 0.98 |
| | | 2 | 0.90 | 0.92 | 0.97 |
| | | 3 | 0.93 | 0.94 | 0.98 |
| | | 4 | 0.87 | 0.91 | 0.98 |
| | | 5 | 0.92 | 0.93 | 0.97 |
| Test set | LGG | 1 | 0.88 | 0.90 | 0.99 |
| | | 2 | 0.84 | 0.89 | 0.98 |
| | | 3 | 0.80 | 0.88 | 0.98 |
| | | 4 | 0.87 | 0.90 | 0.99 |
| | | 5 | 0.80 | 0.87 | 0.98 |
| | HGG | 1 | 0.79 | 0.84 | 0.97 |
| | | 2 | 0.83 | 0.87 | 0.98 |
| | | 3 | 0.80 | 0.85 | 0.97 |
| | | 4 | 0.77 | 0.83 | 0.98 |
| | | 5 | 0.79 | 0.84 | 0.97 |

**Table 4. Performance Comparison and Analysis of Various Network Models.**

| Model | DSC | Recall | Precision |
| --- | --- | --- | --- |
| 3D U-Net model | 0.7512 | 0.7064 | 0.7745 |
| 3D U-Ne+ASPP model | 0.7811 | 0.7244 | 0.7813 |
| Improved 3D U-Net Model | 0.8133 | 0.7769 | 0.8233 |

To assess the impact of different loss functions on glioma segmentation accuracy, three experimental setups were devised using the enhanced 3D U-Net glioma segmentation framework. Scheme 1 employed the Dice loss function; Scheme 2 utilized the Focal loss function; and Scheme 3, the method of this study, combined both the Dice and Focal loss functions. These approaches were applied to segment Whole Tumor (WT), including necrotic and non-enhanced tumors, edema areas, and enhanced tumor areas; Tumor Core (TC), comprising necrotic and non-enhanced tumors, and

enhanced tumor areas; and Enhanced Tumor (ET). The changes in DSC, Recall, and Precision metrics were analyzed across these categories. The segmentation effectiveness of this method in different tumor regions and the corresponding results are displayed in Table 5.

The analysis of Table 5 indicates that the choice of loss function significantly influences glioma segmentation. Utilizing the enhanced 3D U-Net glioma segmentation structure, this study adopts a combined Dice and Focal loss function approach for segmenting glioma in three distinct regions: WT, TC, and ET. The evaluation metrics - DSC, Recall, and Precision - for this method surpass those of the other two schemes. Specifically, in the WT segmentation of the complete tumor region, the DSC, Recall, and Precision values are 0.9168, 0.9426, and 0.9375, respectively. These are 6.47%, 5.65%, 4.71%, and 4.14%; 4.6%, and 4.21% higher than the corresponding values of the other two schemes. For the TC segmentation of the core tumor area, the values are 0.8954, 0.9014, and 0.9369, showing increases of 9.11%, 7.7%, 7.29%, 5.15%, 4.03%, and 3.34% compared to the other schemes. In the ET segmentation of the enhanced tumor region, the values are 0.8674, 0.9045, and 0.9011, outperforming the other schemes by 8.48%, 7.29%, 10.34%, 8.82%, 10.23%, and 6.66%. This method, integrating Dice and Focal loss functions, not only addresses class imbalance in brain glioma segmentation but also enhances learning of hard-to-classify samples, improving recognition of voxels prone to classification errors and significantly enhancing brain glioma segmentation.

To ensure the applicability of the proposed model, the BratS2020 dataset was used as the test set, and 10 samples of 3D MRI glioma cases were selected. FLAIR modal images have good contrast for observing the boundaries of gliomas and surrounding normal brain tissue. Gliomas often have irregular boundaries and blurry areas between them and surrounding normal brain tissue. FLAIR images can help doctors observe these boundaries more clearly, thereby aiding in accurate segmentation of lesions. Therefore, FLAIR modality MRI images were selected for testing. Images of the 10 samples are presented in Fig 5. To assess the segmentation effectiveness of the proposed method, the highly efficient three-dimensional residual neural network segmentation approach from literature [12] was utilized as a comparative method. This approach segmented FLAIR MRI images of the 3D-MRI brain glioma samples alongside the proposed method, with results compared to manual annotations. Differences in brain glioma segmentation outcomes were analyzed, with experimental results showcased in Fig 6, 7, and 8.

From the analysis of Fig 6, 7 and 8, it is evident that FLAIR mode MRI segmentation in this study accurately identifies the complete tumor region WT, core tumor region TC, and enhanced tumor region ET in the MRI samples. The enhanced tumor region ET is marked in red, while the core tumor region TC is indicated by a combination of red and blue. The complete tumor region WT encompasses areas labeled in red, blue, and green. In this study's MRI samples, the brain glioma regions for the 1st, 3rd, 7th, and 9th cases were located in the left hemisphere; for the 2nd case, above the right hemisphere; for the 4th, 5th, 8th, and 10th cases, in the right hemisphere; and for the 6th case, in the lower part of the right

**Table 5. Analysis of the influence of the selection of loss function on the segmentation accuracy of glioma.**

| Evaluation metrics | Tumor area | Segmentation scheme | | |
| --- | --- | --- | --- | --- |
| | | Scheme 1 | Scheme 2 | proposed method |
| DSC | WT | 0.8521 | 0.8603 | 0.9168 |
| | TC | 0.8043 | 0.8184 | 0.8954 |
| | ET | 0.7826 | 0.7945 | 0.8674 |
| Recall | WT | 0.8955 | 0.9012 | 0.9426 |
| | TC | 0.8285 | 0.8499 | 0.9014 |
| | ET | 0.8011 | 0.8163 | 0.9045 |
| Precision | WT | 0.8915 | 0.8954 | 0.9375 |
| | TC | 0.8966 | 0.9035 | 0.9369 |
| | ET | 0.7988 | 0.8345 | 0.9011 |

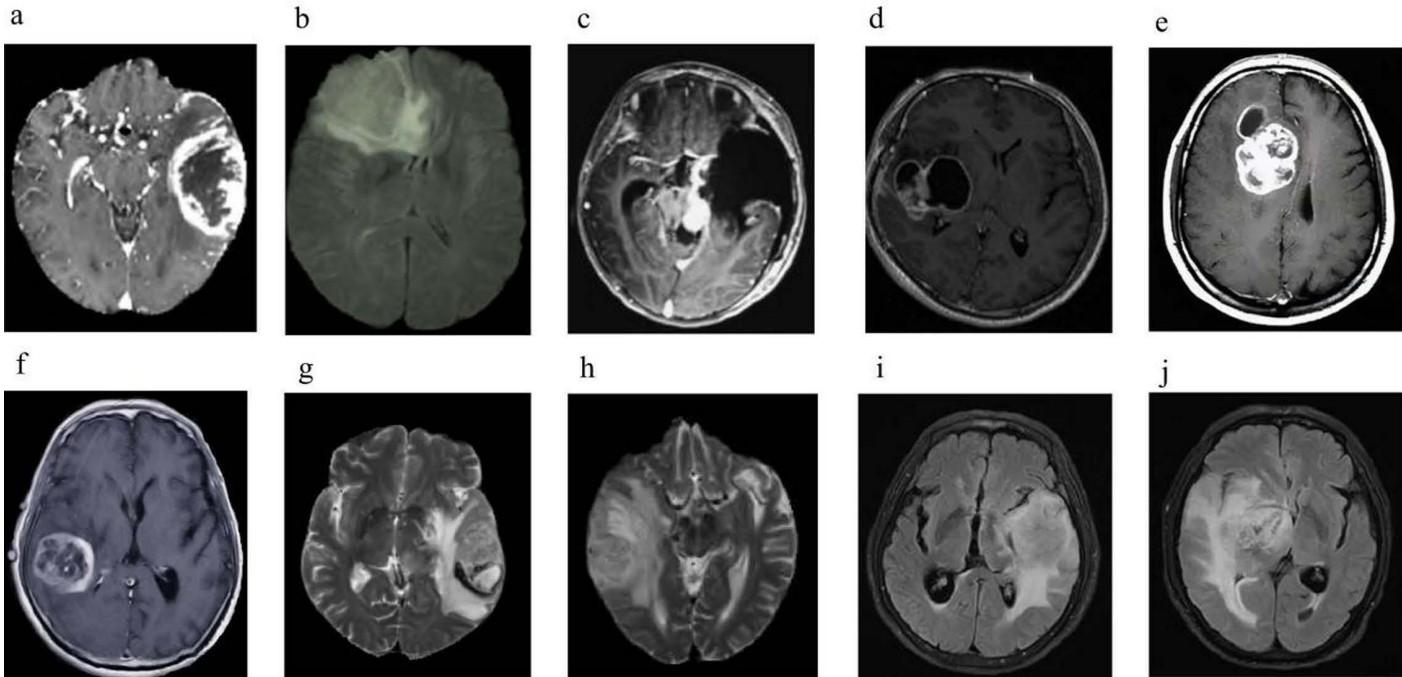

**Fig 5. Original 3D-MRI brain glioma case sample data (a) The 1st 3D-MRI glioma sample (b) The 2nd case of 3D-MRI glioma (c) The 3rd case of 3D-MRI glioma (d) The 4th case of 3D-MRI glioma (e) The 5th case of 3D-MRI glioma (f) The 6th case of 3D-MRI glioma.**

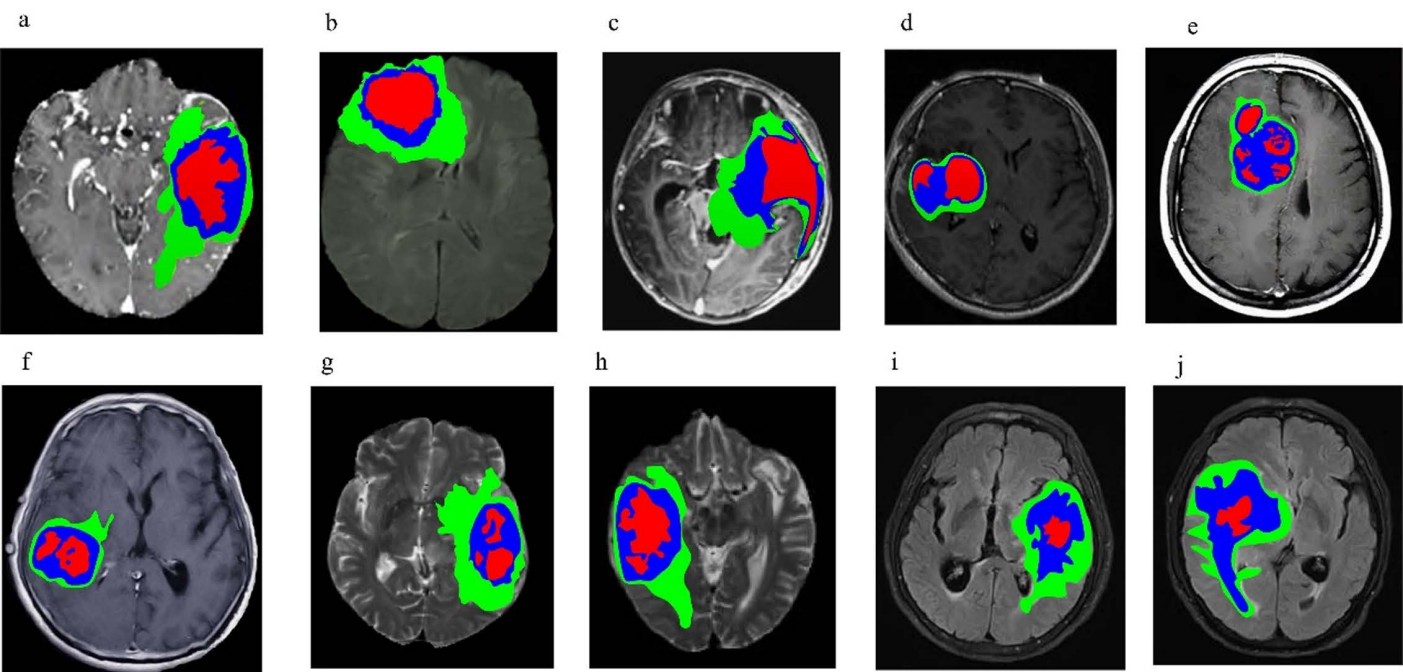

**Fig 6. Sample labeling results of 3D-MRI glioma cases (a) The 1st 3D-MRI glioma sample (b) The 2nd case of 3D-MRI glioma (c) The 3rd case of 3D-MRI glioma (d) The 4th case of 3D-MRI glioma (e) The 5th case of 3D-MRI glioma (f) The 6th case of 3D-MRI glioma.**

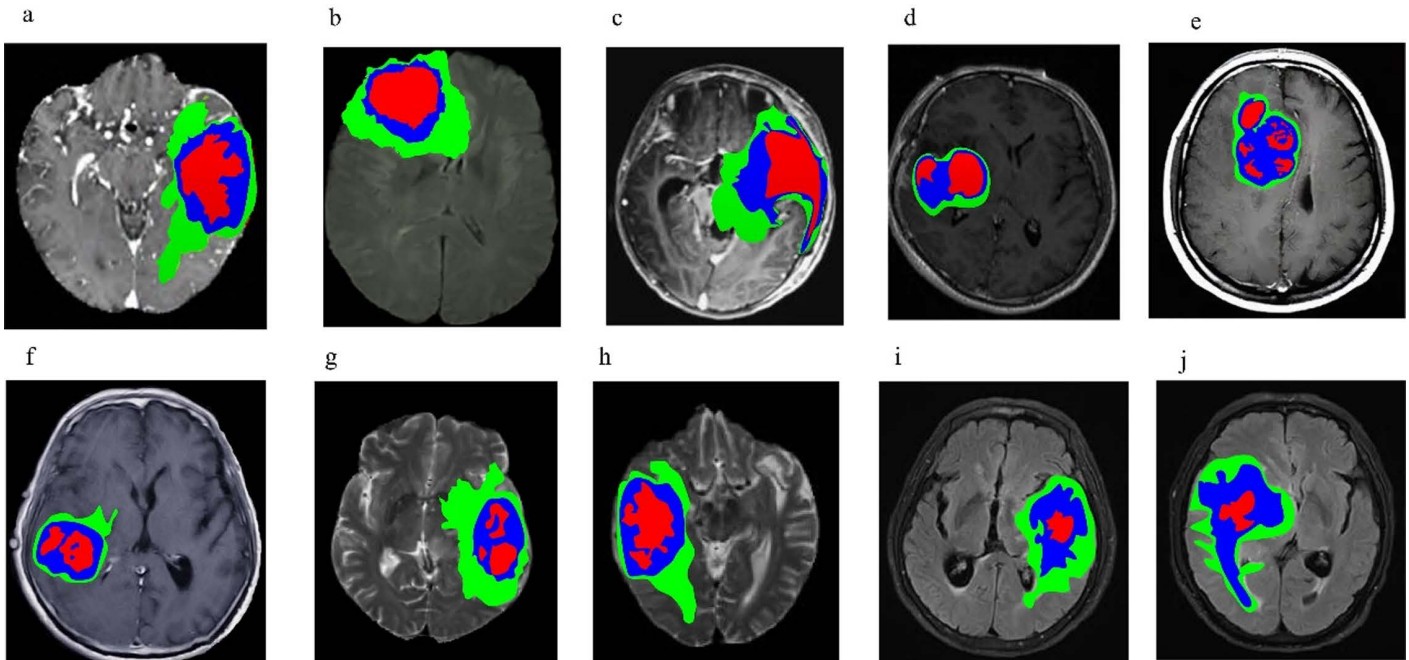

**Fig 7. The segmentation results of 3D-MRI glioma case samples using the method described in this article (a) The 1st 3D-MRI glioma sample (b) The 2nd case of 3D-MRI glioma (c) The 3rd case of 3D-MRI glioma (d) The 4th case of 3D-MRI glioma (e) The 5th case of 3D-MRI glioma (f) The 6th case of 3D-MRI glioma.**

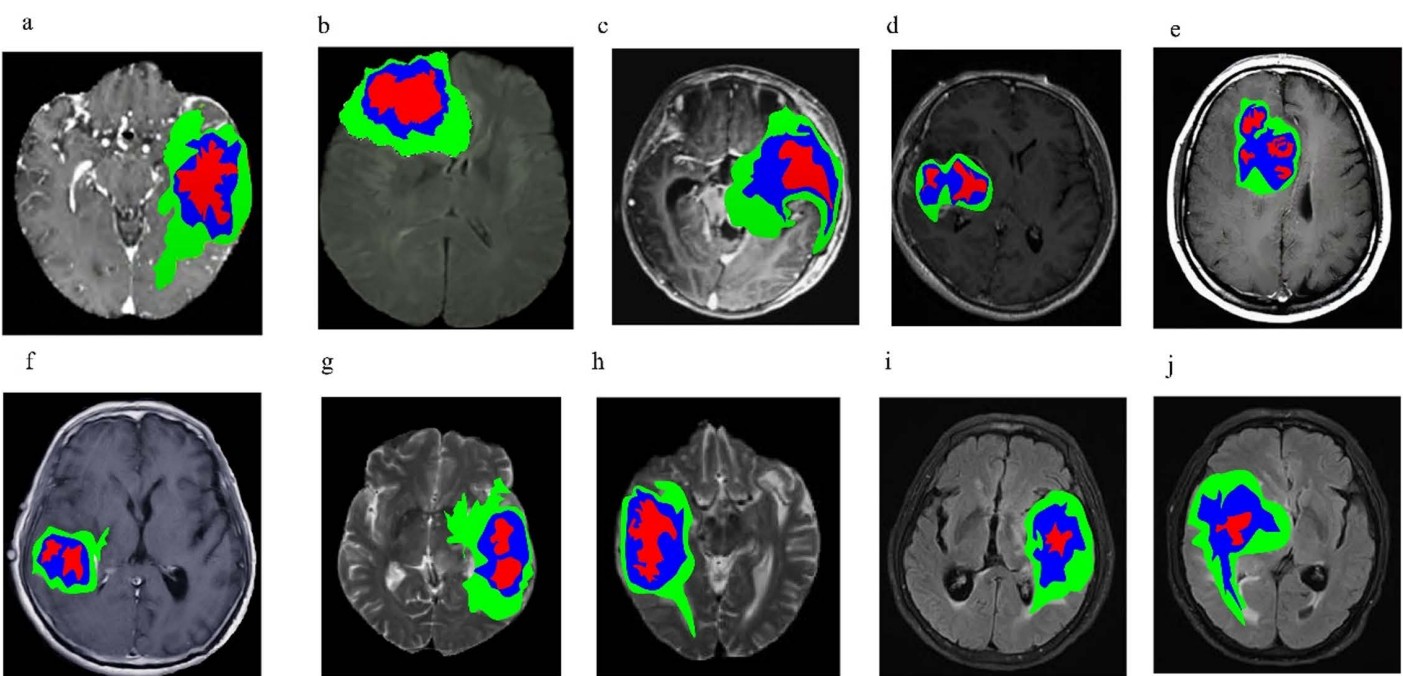

**Fig 8. Sample segmentation results of 3D-MRI glioma cases using comparative methods (a) The 1st 3D-MRI glioma sample (b) The 2nd case of 3D-MRI glioma (c) The 3rd case of 3D-MRI glioma (d) The 4th case of 3D-MRI glioma (e) The 5th case of 3D-MRI glioma (f) The 6th case of 3D-MRI glioma.**

hemisphere. Notably, the enhanced tumor area in the 10 examples was relatively smaller than other tumor areas, making it susceptible to misclassification. Comparing the FLAIR modal MRI segmentation results of the current method with those in reference [12] and the manually annotated results in Fig 7, the enhanced tumor regions segmented by this study are essentially consistent, with only minor differences. However, there is a substantial discrepancy between the method in literature [12] and manual annotation, especially as the enhanced tumor regions are relatively smaller and thus more challenging to segment accurately. The results demonstrate that the method presented in this paper can achieve precise segmentation of brain glioma via 3D-MRI.

On the basis of the above, in order to quantify the segmentation performance of each method, the HD95 distance index is now used to measure the segmentation results of each method on the 10 FLAIR modal image samples mentioned above. The results are shown in Fig 9.

According to the results in Fig 9, using the method proposed in this paper to segment 10 FLAIR modal image samples, the HD95 distance index remained below 0.20. However, after using the method in reference [12] for segmentation, the HD95 distance index in each FLAIR modal image sample reached 0.45. Therefore, comparing the HD95 distance index results of the two methods, it can be concluded that the difference between the predicted segmentation boundary and the real boundary in this paper's method is small, which can better capture the contour of glioma and has high segmentation accuracy. This is because the method proposed in this article uses 3D U-Net as the basic structure, introduces an atrous spatial pyramid pooling structure and a multi-scale fusion attention module to improve the processing of the 3D U-Net network, achieving accurate capture of contextual information at different scales in

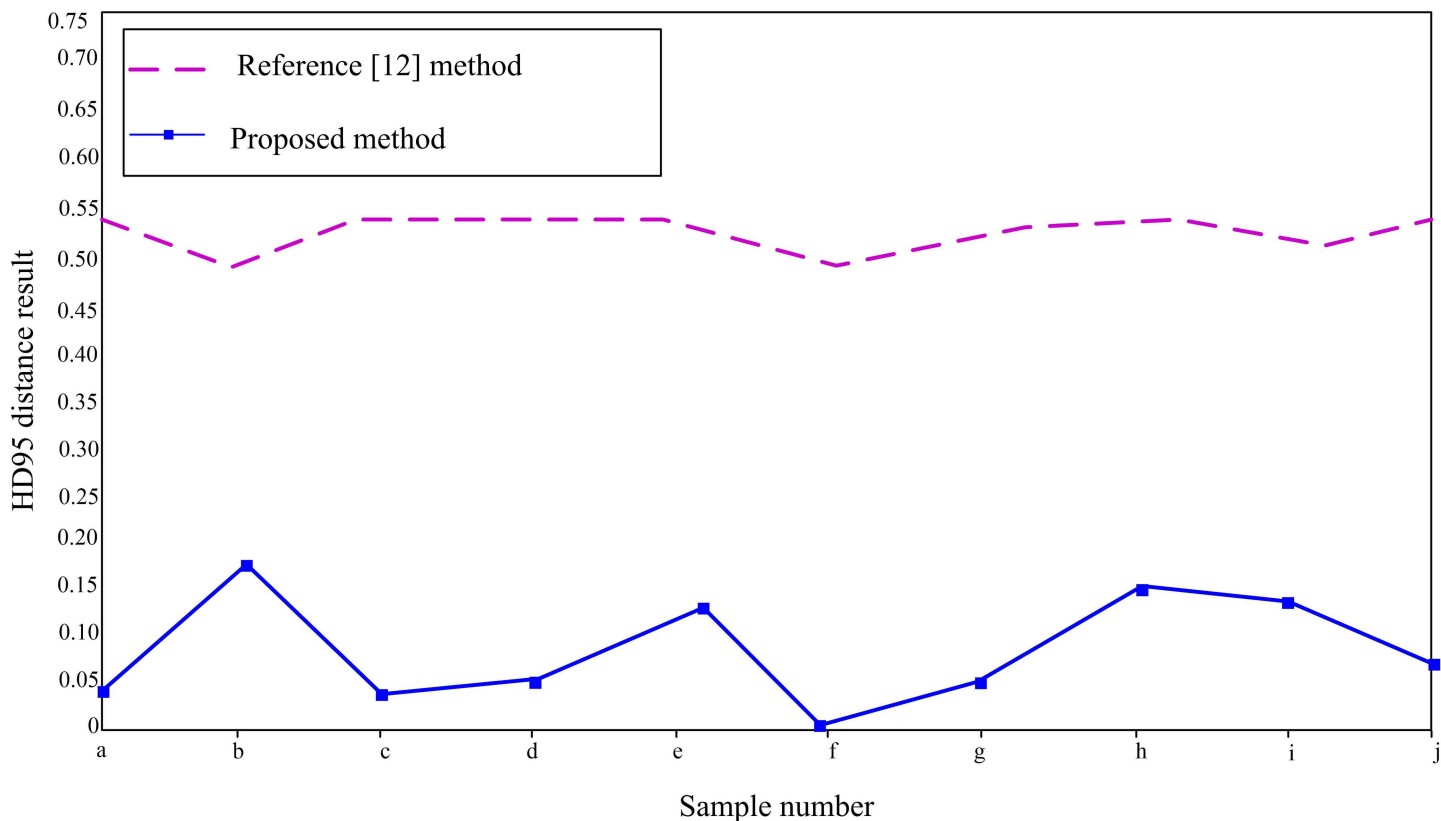

**Fig 9. HD95 distance results.**

brain glioma MRI while enhancing the detailed information of the glioma and suppressing useless background information, thereby improving the performance and effectiveness of segmentation and achieving accurate segmentation of glioma regions.

Based on the above, in order to further verify the performance of the model in boundary delineation and noise suppression, the first and second samples were selected as test samples, and an analysis of segmentation errors was conducted on the red marked enhanced tumor area ET. Under 30dB noise interference, the segmentation error of the proposed model for enhancing tumor area ET in two samples was statistically analyzed, and the expected error was set to 50Pixel The segmentation error results of the proposed model are shown in Fig 10.

According to Fig 10, the proposed method was used to segment the enhanced tumor area ET in the first and second samples, with the highest segmentation error range of 30Pixel to 40Pixel and 18Pixel to 27Pixel, both within the expected range. This indicates that the proposed model has good performance in boundary delineation and noise suppression, with a small error range, ensuring good 3D MRI glioma segmentation results.

To further analyze the computational complexity of the proposed method, the methods in references [11–14], and [15] were used as comparative methods, and time was used to measure the computational complexity. The time complexity can reflect the upward trend of algorithm execution time with increasing input size. Therefore, based on the BraTS2023 public training dataset, 210 samples of advanced glioblastoma (HGG) were used for testing. Six methods were used to segment samples with gradually increasing data size from 10 to 210, measured by linear time complexity. The results are shown in Fig 11.

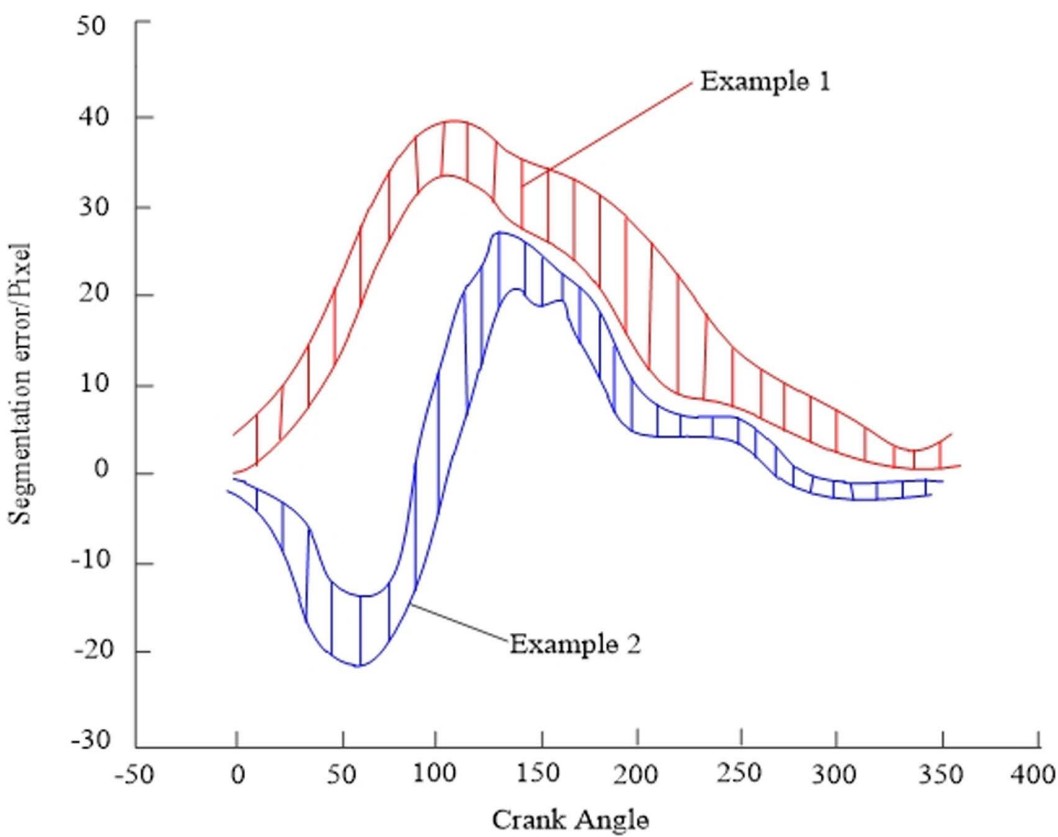

**Fig 10. Segmentation Error Results.**

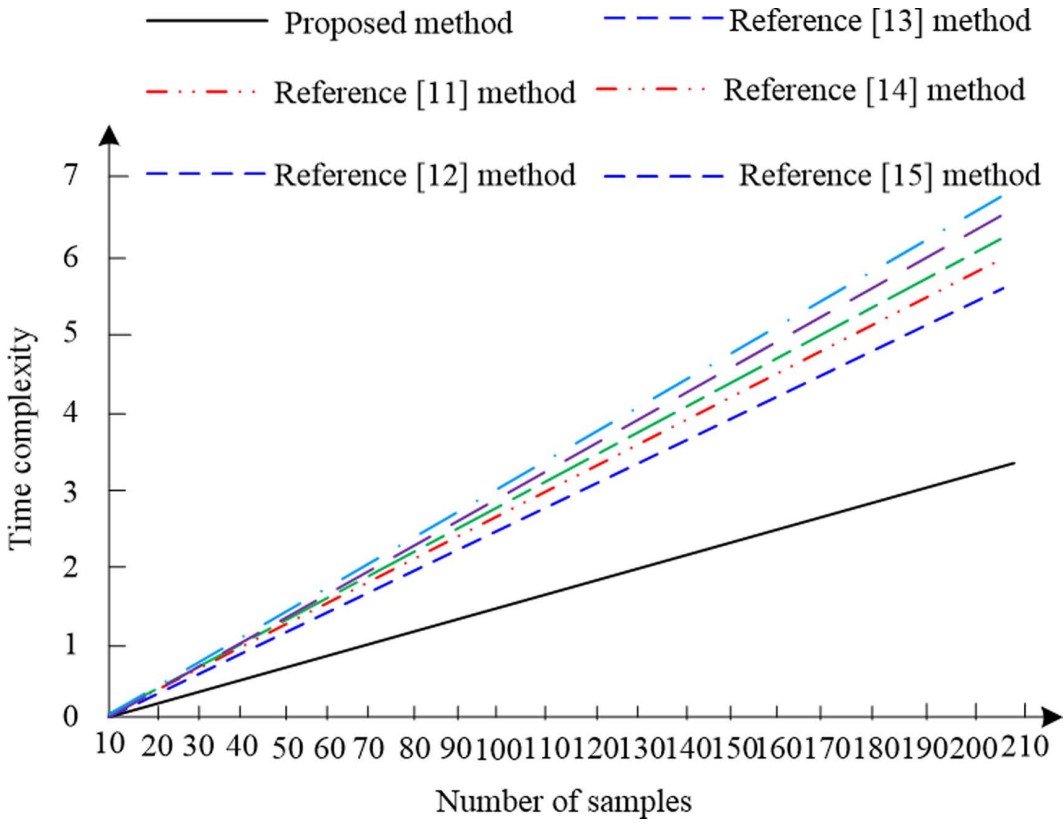

**Fig 11. Linear time complexity results.**

According to the results in Fig 11, using six methods for access control, the linear time complexity results show an upward trend. As the data size increases, the execution time of the methods also increases linearly. However, upon comparison, it can be found that using the proposed method for segmentation results in a lower trend in linear time complexity compared to the other five literature methods. This indicates that the proposed method requires less time and is more efficient when processing data of the same scale. This advantage may stem from the proposed method being more optimized in design, utilizing parallel processing techniques to improve efficiency and resulting in lower computational complexity.

## 5 Discussion

The innovative contribution of this study lies in improving the ability of the model to distinguish challenging samples. By introducing an atrous spatial pyramid pooling structure and a multi-scale fusion attention module, the 3D U-Net network is improved to accurately capture contextual information of different scales in glioma MRI, while enhancing the detailed information of glioma and suppressing useless background information. The Dice loss function and Focal loss function are combined to train the improved 3D U-Net network model, thereby optimizing segmentation accuracy and generalization ability, ultimately improving the accuracy and reliability of diagnosis, and providing a scientific basis for clinical diagnosis and treatment. The test results show that the training loss of the proposed method is only 0.1, with DSC, Recall, and Precision values of 0.7512, 0.7064, and 0.77451, respectively; In whole tumor (WT) segmentation, the Dice similarity coefficient (DSC), recall rate, and accuracy scores were 0.9168, 0.9426, and 0.9375, respectively; For the segmentation of core tumors (TC), these scores are 0.8954, 0.9014, and 0.9369, respectively; In enhanced tumor (ET) segmentation, the DSC,

Recall, and Precision values of this method are 0.8674, 0.9045, and 0.9011, respectively; And the HD95 distance index remains above 0.98, with low time complexity. This indicates that the introduction of atrous spatial pyramid pooling structure and multi-scale fusion attention module to improve the processing of 3D U-Net network, and based on the concept of parallel computing to overcome the problem of requiring a large amount of memory and processing power for the model to be computationally intensive, effectively enhances the 3D U-Net segmentation model and exhibits excellent performance.

Although the proposed method overcomes the problem of large memory and processing power required for computationally intensive models based on the concept of parallel computing, there are still issues of introducing additional overhead and increasing computing resources. Therefore, in the future, research will be conducted towards optimizing model structure and computational efficiency to explore more efficient network architectures, such as lightweight 3D convolutional neural networks, to reduce the consumption of computing resources and memory.

## 6 Conclusion

Using the 3D U-Net network as the foundational structure, this study introduces the ASPP and multi-scale fusion attention module (MFAB) to construct an enhanced glioma segmentation network mode. This model leverages multiple receptive fields of varying sizes to acquire multi-scale contextual information from glioma MRI images, thereby improving the network model's feature extraction capabilities. It also excels in capturing valuable salient features in specific regions while effectively suppressing information irrelevant to glioma segmentation. Additionally, a loss function, combining Dice loss and Focal loss, is devised to efficiently regulate the parameters of the loss function, aiming to enhance the segmentation model's performance. Within the Pytorch framework, 225 random glioma samples from the BraTS2023 public training set were used to train the segmentation network model, and its performance was evaluated using 60 test samples. The experimental results reveal that:

(1) The enhanced 3D U-Net segmentation model exhibits exceptional performance, achieving a training loss of only 0.1, with DSC, Recall, and Precision indices surpassing those of both the standard 3D U-Net and the 3D U-Net+ASPP network models.

(2) The use of a loss function that combines Dice loss function and Focal loss function significantly improves the performance of glioma segmentation.

(3) This method successfully segments 3D-MRI glioma, accurately identifying the complete tumor area (WT), core tumor area (TC), and enhanced tumor area (ET), demonstrating high segmentation accuracy.

## Supporting information

**S1 Data. Partial Data.**
(ZIP)

## Author contributions

**Conceptualization:** Tingting Wang, Tong Wu, Defu Yang, Ying Xu, Tong Jiang, Hengjiao Wang, Qi Chen.

**Data curation:** Tingting Wang, Tong Wu, Defu Yang, Ying Xu, Tong Jiang, Hengjiao Wang, Qi Chen, Shengnan Xu.

**Formal analysis:** Shengnan Xu.

**Funding acquisition:** Ying Xu.

**Investigation:** Dongyang Lv.

**Methodology:** Tingting Wang, Ying Xu, Dongyang Lv, Tong Jiang.

**Project administration:** Ying Xu, Dongyang Lv.

 

**Writing – original draft:** Tingting Wang, Tong Wu, Qi Chen.

**Writing – review & editing:** Tingting Wang, Tong Wu, Shengnan Xu.

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
