## [Decision Letter · Decision Letter 0]

Nov 18 2024

Dear Dr. Wang,

Thank you for submitting your manuscript to PLOS ONE. After careful consideration, we feel that it has merit but does not fully meet PLOS ONE’s publication criteria as it currently stands. Therefore, we invite you to submit a revised version of the manuscript that addresses the points raised during the review process.

We look forward to receiving your revised manuscript.

Kind regards,

Sarada Prasad Dakua

Academic Editor

PLOS ONE

Journal Requirements:

3. Please note that PLOS ONE has specific guidelines on code sharing for submissions in which author-generated code underpins the findings in the manuscript. In these cases, we expect all author-generated code to be made available without restrictions upon publication of the work. Please review our guidelines at https://journals.plos.org/plosone/s/materials-and-software-sharing#loc-sharing-code and ensure that your code is shared in a way that follows best practice and facilitates reproducibility and reuse."

4. Please note that funding information should not appear in any section or other areas of your manuscript. We will only publish funding information present in the Funding Statement section of the online submission form. Please remove any funding-related text from the manuscript.

5. We note that the grant information you provided in the ‘Funding Information’ and ‘Financial Disclosure’ sections do not match. When you resubmit, please ensure that you provide the correct grant numbers for the awards you received for your study in the ‘Funding Information’ section.

6. PLOS requires an ORCID iD for the corresponding author in Editorial Manager on papers submitted after December 6th, 2016. Please ensure that you have an ORCID iD and that it is validated in Editorial Manager. To do this, go to ‘Update my Information’ (in the upper left-hand corner of the main menu), and click on the Fetch/Validate link next to the ORCID field. This will take you to the ORCID site and allow you to create a new iD or authenticate a pre-existing iD in Editorial Manager.

Additional Editor Comments:

Please work on the novelty part and address the required comments made by the reviewers.

Reviewers' comments:

Reviewer's Responses to Questions

**Comments to the Author**

1. Is the manuscript technically sound, and do the data support the conclusions?

Reviewer #1: Yes

Reviewer #2: Yes

2. Has the statistical analysis been performed appropriately and rigorously?

Reviewer #1: N/A

Reviewer #2: N/A

3. Have the authors made all data underlying the findings in their manuscript fully available?

Reviewer #1: Yes

Reviewer #2: Yes

4. Is the manuscript presented in an intelligible fashion and written in standard English?

Reviewer #1: Yes

Reviewer #2: Yes

Reviewer #1: The proposed 3D U-Net architecture seems to have some limitations: Firstly, novelty remains a concern; thus, the authors need to present the contributions clearly in the revised manuscript.

1. Due to the high number of parameters, it can easily overfit to the training data, especially if the dataset is small or not diverse enough.

2. The model seems to be computationally intensive, requiring significant memory and processing power, which can be a barrier for some researchers or clinics. Is it possible to use the concept of parallel computation to overcome this? The authors could discuss this by referring the below papers:

“Real-time Automated Image Segmentation Technique for Cerebral Aneurysm on Reconfigurable System-On-Chip,” Journal of Computational Science, Elsevier, vol. 27, pp 35-45, 2018.

“Lattice-Boltzmann Interactive Blood Flow Simulation Pipeline,” International Journal of Computer Assisted Radiology and Surgery, Springer, vol.15, pp. 629-639, 2020.

“Zynq SoC based Acceleration of the Lattice Boltzmann Method,” Concurrency and Computation: Practice and Experience, Wiley, col. 31, issue 17, 2019.

"Heterogeneous System-on-Chip based Lattice- Boltzmann Visual Simulation System,” Systems Journal, IEEE, vol. 14, no. 2, pp. 1592-1601, 2020

3. In cases where the glioma regions are much smaller compared to healthy tissue, the model may have difficulty learning the minority class effectively, leading to poor segmentation performance.

The performance may be adversely affected by noise and artifacts in the MRI images, which are common in clinical settings. The authors need to discuss if a pre-processing using stocastic resonance theory can be of help. The authors could refer the bew studies while discussing this: "Development of a Cerebral Aneurysm Segmentation Method to Prevent Sentinel Hemorrhage," Network Modeling Analysis in Health Informatics and Bioinformatics, Springer, vol. 12, no. 18, pp. 1-14, 2023.

"Toward Computing Cross-Modality Symmetric Non-Rigid Medical Image Registration," IEEE Access, vol. 10, pp. 24528-24539, 2022,

"Moving Object Tracking in Clinical Scenarios: Application to Cardiac Surgery and Cerebral Aneurysm Clipping,” International Journal of Computer Assisted Radiology and Surgery, Springer, vol. 14, no. 12, pp. 2165-2176, 2019.

“A PCA based Approach for Brain Aneurysm Segmentation,” Journal of Multi Dimensional Systems and Signal Processing, Springer, vol. 29, pp. 257-277, 2018.

4. The authors are encourage to include the potential limitations of the paper.

5. Please discuss the present computational complexity of the model.

Reviewer #2: Authors have exaggerated their claims and much of the claims have already been done in literature:

1. "hyperparametric loss function by combining the Dice loss and Focal loss functions" This is a well known combo loss in the literature.

2. Use of pyramid scene parsing and ASPP has been popular for medical image segementation:

a. A lightweight neural network with multiscale feature enhancement for liver CT segmentation

b. Dense-PSP-UNet: A neural network for fast inference liver ultrasound segmentation

3. Authors claim that registration is part of their method. However, no details of registration have been shared in the the methods section.

4. What's the distinction between MFAB and fusion module? Network figure has to be redesigned and explanation should be made clear.

5. Please dont exaggerate loss as super parameter or hyperparametric.

6. training loss is not interesting why not share validation and test losses for Tables 1 and 2.

7. HD distance as a metrics is skipped in the results section.

8. Lack of comparison with literature. At least 5-7 methods should be implemented and compared with the proposed method.

9. Discussion is missing from the work.

10. Following works should be cited to encourage the use of DL/CNN/UNet for brain tumor segmentation:

a. Towards developing a lightweight neural network for liver CT segmentation.

b. Neural network-based fast liver ultrasound image segmentation

c. Unveiling the future of breast cancer assessment: a critical review on generative adversarial networks in elastography ultrasound

d. Advancements in Deep Learning for B-Mode Ultrasound Segmentation: A Comprehensive Review

e. Estimating age and gender from electrocardiogram signals: A comprehensive review of the past decade

f. Practical utility of liver segmentation methods in clinical surgeries and interventions

**Do you want your identity to be public for this peer review?** For information about this choice, including consent withdrawal, please see our Privacy Policy

Reviewer #1: No

Reviewer #2: No

---

## [Author Response · Author response to Decision Letter 1]

7 Nov 2024

Dear Editors and Reviewers:

Thank you for your letter and for the reviewers’ comments concerning our manuscript entitled “3D-MRI Brain Glioma Intelligent Segmentation Based on Improved 3D U-Net Network” (ID: PONE-D-24-42605). Those comments are all valuable and very helpful for revising and improving our paper, as well as the important guiding significance to our researches. We have studied comments carefully and have made correction which we hope meet with approval.

Responds to the reviewer’s comments:

Reviewer #1:

Thank you very much for your professional review of our articles. In the revised version, your suggestions are highlighted in red. According to your suggestions, we have made a lot of modifications to the previous draft, and the specific modifications are as follows:

The proposed 3D U-Net architecture seems to have some limitations: Firstly, novelty remains a concern; thus, the authors need to present the contributions clearly in the revised manuscript.

1. Due to the high number of parameters, it can easily overfit to the training data, especially if the dataset is small or not diverse enough.

Reply: Based on the issues pointed out above, in order to solve the problem of imbalanced datasets with small or insufficient diversity, the article uses a loss function composed of Dice loss function and Focal loss function to train an improved 3D U-Net network model, in order to improve the model's generalization ability. The specific description begins with section 3.5 as follows:

This function comprises both the Dice loss and Focal loss functions. Train the improved 3D U-Net network model with the loss function as the optimization objective to comprehensively consider the accuracy of prediction results and class imbalance issues, thereby improving the model's generalization ability and ensuring the accuracy of 3D MRI segmentation of gliomas. The Dice loss function, which is the value obtained by subtracting the Dice coefficient from 1, transforms the voxel level labeling problem into a problem of minimizing inter class distance to learn the distribution of classes, and to some extent solves the problem of class imbalance; The Focal loss function changes the sample weights to make the model more inclined to focus on difficult to classify samples during training, in order to better learn voxels with classification errors.

2. The model seems to be computationally intensive, requiring significant memory and processing power, which can be a barrier for some researchers or clinics. Is it possible to use the concept of parallel computation to overcome this? The authors could discuss this by referring the below papers:

“Real-time Automated Image Segmentation Technique for Cerebral Aneurysm on Reconfigurable System-On-Chip,” Journal of Computational Science, Elsevier, vol. 27, pp 35-45, 2018.

“Lattice-Boltzmann Interactive Blood Flow Simulation Pipeline,” International Journal of Computer Assisted Radiology and Surgery, Springer, vol.15, pp. 629-639, 2020.

“Zynq SoC based Acceleration of the Lattice Boltzmann Method,” Concurrency and Computation: Practice and Experience, Wiley, col. 31, issue 17, 2019.

"Heterogeneous System-on-Chip based Lattice- Boltzmann Visual Simulation System,” Systems Journal, IEEE, vol. 14, no. 2, pp. 1592-1601, 2020

Reply: Based on the suggestions, a search was conducted on the provided literature, and in terms of processing efficiency, corresponding descriptions were made in the article according to the concept of parallel computing, as follows:

And to cope with the large amount of memory and processing power required for computationally intensive models, the concept of parallel computing is used to overcome this problem. The hollow space pyramid pooling structure introduced in the 3D U-Net network and the network structure of the multi-scale fusion attention module are run separately on different GPUs, and data transmission is carried out through high-speed communication links between the hollow space pyramid pooling structure and the network structure of the multi-scale fusion attention module to ensure communication and data synchronization between different GPUs. And the tasks in the multi-scale fusion attention module can also be decomposed into smaller tasks, that is, the calculation of different scale features can be performed in parallel on different computing units, and then the fusion operation can be performed. Based on this, to some extent, the problem of high memory and processing power requirements brought by the model can be overcome, and overall efficiency can be improved.

3. In cases where the glioma regions are much smaller compared to healthy tissue, the model may have difficulty learning the minority class effectively, leading to poor segmentation performance.

The performance may be adversely affected by noise and artifacts in the MRI images, which are common in clinical settings. The authors need to discuss if a pre-processing using stocastic resonance theory can be of help. The authors could refer the bew studies while discussing this: "Development of a Cerebral Aneurysm Segmentation Method to Prevent Sentinel Hemorrhage," Network Modeling Analysis in Health Informatics and Bioinformatics, Springer, vol. 12, no. 18, pp. 1-14, 2023.

"Toward Computing Cross-Modality Symmetric Non-Rigid Medical Image Registration," IEEE Access, vol. 10, pp. 24528-24539, 2022,

"Moving Object Tracking in Clinical Scenarios: Application to Cardiac Surgery and Cerebral Aneurysm Clipping,” International Journal of Computer Assisted Radiology and Surgery, Springer, vol. 14, no. 12, pp. 2165-2176, 2019.

“A PCA based Approach for Brain Aneurysm Segmentation,” Journal of Multi Dimensional Systems and Signal Processing, Springer, vol. 29, pp. 257-277, 2018.

Reply: Based on the suggestions, the provided literature has been referenced, and in section 2.2, preprocessing based on stochastic resonance theory has been added to avoid the adverse effects of noise on subsequent segmentation. The specific supplementary content is as follows:

Based on the above corrections, in order to further avoid the impact of noise on the subsequent segmentation performance of MRI images, we now proceed with processing according to the theory of stochastic resonance. Random resonance is a nonlinear phenomenon, and under certain conditions, the addition of noise can enhance the detection ability of weak signals. In MRI images, although noise is often considered interference, it is possible to transform noise into a factor that helps improve image quality by utilizing stochastic resonance theory. From the perspective of signal processing, the signal in MRI images can be seen as a mixed signal consisting of target tissue information (useful signal) and noise and artifacts (interference signal). The theory of stochastic resonance can enhance useful signals in the presence of noise by adjusting certain parameters of the system. The specific implementation process is as follows:

Firstly, a method based on local statistical information is used to estimate the noise level in MRI images, as shown below:

5

In the formula, represents the noise intensity; is the adjustment coefficient, which can take values between 0.1-10; is the noise variance.

Secondly, the modified and normalized data mentioned above is used as the amplitude of the input MRI image signal, which is then fed into a bistable system. The dynamic equation can be expressed as:

6

In the formula, is the state variable of the system; is the angular frequency of the input signal, with an initial value of 1, and then adjusted according to the processing result; is Gaussian white noise.

Then, numerical methods are used to solve the dynamic equations of the bistable system, and the output signal after stochastic resonance processing is obtained. For the grayscale value of each pixel, the value of the system's state variable is gradually calculated according to a set time step (such as ). After a certain period of time (such as time units), the stable value of is taken as the processed grayscale value of the pixel, and the corresponding output signal is obtained.

Finally, the one-dimensional signal processed by stochastic resonance is converted back into a two-dimensional image. Evaluate the processed image using the Peak Signal to Noise Ratio (PSNR) image quality assessment metric. If the evaluation indicators do not improve, the parameters of the stochastic resonance system (such as noise intensity , angular frequency ) can be adjusted and reprocessed until satisfactory image quality improvement is achieved.

4. The authors are encourage to include the potential limitations of the paper.

Reply: According to the suggestion, a description of the limitations of the proposed method has been added to the discussion section in Section 5, as follows:

Although the proposed method overcomes the problem of large memory and processing power required for computationally intensive models based on the concept of parallel computing, there are still issues of introducing additional overhead and increasing computing resources.

5. Please discuss the present computational complexity of the model.

Reply: According to the suggestion, an analysis on computational complexity has been added in Section 4, as follows:

To further analyze the computational complexity of the proposed method, the methods in references [11], [12], [13], [14], and [15] were used as comparative methods, and time was used to measure the computational complexity. The time complexity can reflect the upward trend of algorithm execution time with increasing input size. Therefore, based on the BraTS2018 public training dataset, 210 samples of advanced glioblastoma (HGG) were used for testing. Six methods were used to segment samples with gradually increasing data size from 10 to 210, measured by linear time complexity. The results are shown in Figure 11.

Figure 11 Linear time complexity results

According to the results in Figure 11, using six methods for access control, the linear time complexity results show an upward trend. As the data size increases, the execution time of the methods also increases linearly. However, upon comparison, it can be found that using the proposed method for segmentation results in a lower trend in linear time complexity compared to the other five literature methods. This indicates that the proposed method requires less time and is more efficient when processing data of the same scale. This advantage may stem from the proposed method being more optimized in design, utilizing parallel processing techniques to improve efficiency and resulting in lower computational complexity.

Reviewer #2:

We sincerely thank reviewers for their valuable comments, which we use to improve the quality of our manuscripts. The reviewer's comments are listed below in bold, with specific questions numbered. Our replies are given in green text.

1. "hyperparametric loss function by combining the Dice loss and Focal loss functions" This is a well known combo loss in the literature.

Reply: The pointed out issue has been checked in the article, and its innovation does not lie in combining Dice loss and focus loss functions. Instead, a hollow space pyramid pooling structure and a multi-scale fusion attention module are introduced to improve the processing of the 3D U-Net network. While accurately capturing contextual information of different scales in glioma MRI, the detailed information of glioma is enhanced, and useless background information is suppressed to improve segmentation performance. Based on this, in order to improve the segmentation accuracy and generalization ability of the model, a loss function combining Dice loss function and Focal loss function is used to train the improved model, enhance its learning ability for difficult to distinguish samples, and further improve segmentation performance.

2. Use of pyramid scene parsing and ASPP has been popular for medical image segementation:

a. A lightweight neural network with multiscale feature enhancement for liver CT segmentation

b. Dense-PSP-UNet: A neural network for fast inference liver ultrasound segmentation

Reply: Based on the pointed out issues, the proposed method introduces the advantages of the hollow space pyramid pooling structure and multi-scale fusion attention module into the 3D U-Net network to improve the 3D U-Net network. The two structures complement each other, making significant progress in capturing multi-scale information, enhancing feature extraction ability, and improving segmentation accuracy. This improves the effectiveness of the 3D U-Net network in 3D MRI glioma segmentation and provides strong support for accurate diagnosis and treatment of gliomas.

3. Authors claim that registration is part of their method. However, no details of registration have been shared in the the methods section.

Reply: Based on the identified issues, corresponding implementation process content has been added in section 2.1 of the article, as follows:

During the registration process, the deformation parameters of the input image were continuously adjusted to minimize the difference between them and the reference image. The specific implementation process is described as follows:

During the registration process, the mean square error (MSE) is used to measure the similarity between the input image and the reference image, with the goal of minimizing this MSE value. The mean square error is expressed as follows:

1

In the formula, is the input image; is the reference image; is the total number of pixels.

Registration continuously adjusts the deformation parameters of the input image to minimize the difference between it and the reference image. This deformation parameter is usually represented by a displacement vector field, namely deformation field , which describes the deformation mapping from the input image to the reference image, that is, each pixel (or voxel) in each direction has a corresponding displacement vector. represents as follows:

2

To prevent excessive deformation, a regularization term is added during the registration process to constrain the smoothness of the deformation field. The regularization term is represented as follows:

3

In the equation, is the gradient of the displacement vector; is the modulus of the gradient vector.

Then, through iterative optimization, continuously adjust the deformation parameters. In each iteration, the similarity measure and regularization term will be calculated based on the current deformation parameters, and the deformation parameters will be updated using gradient descent until the MSE value of the similarity measure reaches the minimum and satisfies the constraint of formula (3), ending the iteration and completing the registration.

4. What's the distinction between MFAB and fusion module? Network figure has to be redesigned and explanation should be made clear.

Reply: The difference between MFAB and fusion module is that MFAB emphasizes channel attention mechanism, which enhances or suppresses feature maps by learning the importance of each feature map channel. However, typical fusion modules do not have this explicit channel attention mechanism.

And MFAB pays special attention to the integration of multi-scale features, which helps capture various information in images. And some fusion modules may only focus on feature integration at specific scales.

And based on the suggestions, the structure of the multi-scale fusion attention module was redrawn.

5. Please dont exaggerate loss as super parameter or hyperparametric.

Reply: According to the suggestion, unify the hyperparameter loss function into a loss function.

6. training loss is not interesting why not share validation and test losses for Tables 1 and 2.

Reply: Table 2 shows the robustness results of each model to interference on the test dataset. Therefore, according to the suggestion, Table 1 should be changed to the loss on the validation and test sets, as follows:

The efficacy of the brain glioma segmentation model significantly influences its segmentation accuracy. Utilizing the 3D U-Net network as a foundational structure, this paper introduces the ASPP netwo

---

## [Decision Letter · Decision Letter 1]

Feb 22 2025

Dear Dr. Wang,

Thank you for submitting your manuscript to PLOS ONE. After careful consideration, we feel that it has merit but does not fully meet PLOS ONE’s publication criteria as it currently stands. Therefore, we invite you to submit a revised version of the manuscript that addresses the points raised during the review process.

We look forward to receiving your revised manuscript.

Kind regards,

Tao Peng

Academic Editor

PLOS ONE

**Additional Editor Comments:**

It is the revised version, while there exists several issues, including lack of comprehensive comparisons of SOTAs, lack of statistical significance, and Insufficient Result Analysis.

Reviewers' comments:

Reviewer's Responses to Questions

**Comments to the Author**

Reviewer #3: All comments have been addressed

Reviewer #4: All comments have been addressed

2. Is the manuscript technically sound, and do the data support the conclusions?

Reviewer #3: Partly

Reviewer #4: Partly

3. Has the statistical analysis been performed appropriately and rigorously?

Reviewer #3: No

Reviewer #4: Yes

4. Have the authors made all data underlying the findings in their manuscript fully available?

Reviewer #3: Yes

Reviewer #4: Yes

5. Is the manuscript presented in an intelligible fashion and written in standard English?

Reviewer #3: No

Reviewer #4: Yes

**Reviewer #3:**  This paper resembles a technical report rather than a scientific research article. While the performance results are notable, the following concerns must be addressed before it can be considered for publication.

1.Writing and Grammar:

The overall writing of the manuscript requires improvement. For example, phrases like "Validate the dataset" and "Test dataset" on page 14 are unclear. There are grammar errors and clarity issues throughout the manuscript. Tools such as ChatGPT or other AI-based writing assistants can help enhance the writing quality. Ensure that every sentence in the manuscript is reviewed for grammatical accuracy and clarity.

2.State-of-the-Art (SOTA) Comparison:

To demonstrate that your method is truly state-of-the-art, it should be compared with the latest models on up-to-date datasets, such as those from the BraTS 2023 challenge. Since most SOTA methods no longer compete on older datasets, references to BraTS 2018 are largely outdated. Additionally, the baseline model used in the paper, U-Net with ASPP, is nearly 9 years old and does not reflect the latest advancements in the field.

3.Emphasis on Novelty:

To highlight the novelty of this paper, focus on identifying existing challenges and proposing innovative solutions, rather than combining or stacking existing methods. A clear distinction of how your approach addresses unmet needs in the field will strengthen the paper.

4.Background Information:

Commonly known background concepts, such as the original U-Net architecture, ASPP diagrams, multi-scale fusion attention modules, and formulas for Dice/recall/precision, should be abbreviated or summarized concisely to avoid redundancy.

5.Overfitting and Data Inclusion:

Changing the loss function alone does not adequately address the overfitting problem. A more effective approach would be to incorporate additional public datasets into the training process. This would improve the model's generalizability and robustness.

6.Test Set Usage (Page 14, Lines 454–455):

The statement, "In the test dataset, as the number of training iterations increases, the training loss of each glioma segmentation network model shows a decreasing trend, and the final region stabilizes," raises serious concerns. Did you inadvertently include the test set during training? This practice is entirely unacceptable, as it compromises the validity of your results. If the improvement in Dice score is derived from using the test set during training, this constitutes a form of cheating. The test set must remain completely unseen during training and should only be used for final evaluation.

7.Validation vs. Test Set Results (Table 1):

In Table 1, the loss values for the validation set and the test set are shown to be exactly the same. This is highly unusual and warrants a thorough review. Please double-check your experimental setup and ensure there are no errors in your data partitioning or evaluation process.

**Reviewer #4: ** Expand Comparative Experiments:

Include comparisons with recent high-performing glioma segmentation methods (e.g., those utilizing Transformers or advanced 3D architectures) to comprehensively evaluate the proposed method’s competitiveness.

Test the model on additional datasets (e.g., BraTS2020) and consider incorporating multicenter clinical datasets to enhance the study’s real-world applicability.

Deepen Ablation Studies:

Conduct detailed ablation studies to independently quantify the contributions of the ASPP and MFAB modules, and analyze the effects of varying hyperparameters.

Strengthen Result Analysis:

Include statistical significance testing and error range reporting to ensure the scientific rigor and reliability of the results.

Provide additional visualizations (e.g., segmentation error maps) and analyze the model’s performance in boundary delineation and noise suppression.

Improve Methodological Novelty:

Consider further optimizing or introducing novel variations to the ASPP and MFAB modules to enhance the originality of the proposed approach.

**Do you want your identity to be public for this peer review?** For information about this choice, including consent withdrawal, please see our Privacy Policy

Reviewer #3: **Yes: ** Hengrui Zhao

Reviewer #4: No

---

## [Author Response · Author response to Decision Letter 2]

13 Mar 2025

Dear Editors and Reviewers:

Thank you for your letter and for the reviewers’ comments concerning our manuscript entitled “3D-MRI Brain Glioma Intelligent Segmentation Based on Improved 3D U-Net Network” (ID: PONE-D-24-42605R1). Those comments are all valuable and very helpful for revising and improving our paper, as well as the important guiding significance to our researches. We have studied comments carefully and have made correction which we hope meet with approval.

---

## [Decision Letter · Decision Letter 2]

Jun 08 2025

Dear Dr. Wang,

Thank you for submitting your manuscript to PLOS ONE. After careful consideration, we feel that it has merit but does not fully meet PLOS ONE’s publication criteria as it currently stands. Therefore, we invite you to submit a revised version of the manuscript that addresses the points raised during the review process.

We look forward to receiving your revised manuscript.

Kind regards,

Tao Peng

Academic Editor

PLOS ONE

Journal Requirements:

Additional Editor Comments:

It is the revised version with better quality, while several minor revisions are needed to handle the residual issues, such as 1) the authors should improve the readability of this submission.

Reviewers' comments:

Reviewer's Responses to Questions

**Comments to the Author**

Reviewer #3: (No Response)

Reviewer #4: All comments have been addressed

2. Is the manuscript technically sound, and do the data support the conclusions?

Reviewer #3: No

Reviewer #4: Yes

3. Has the statistical analysis been performed appropriately and rigorously?

Reviewer #3: No

Reviewer #4: Yes

4. Have the authors made all data underlying the findings in their manuscript fully available?

Reviewer #3: Yes

Reviewer #4: Yes

5. Is the manuscript presented in an intelligible fashion and written in standard English?

Reviewer #3: No

Reviewer #4: Yes

Reviewer #3: I appreciate the authors’ efforts in improving the paper. However, several issues remain unsatisfactory:

1. Grammar and Consistency

While the authors revised "Validate the dataset" in one paragraph, it still appears unchanged in Table 1 and Table 3. Please ensure consistency throughout the paper.

Standardize expressions such as "training set," "validation set," and "test set" across all instances.

"Data set" should be consistently written as "Dataset."

I recommend using an AI proofreading tool, such as DeepSeek or Qwen, to check and correct grammar errors throughout the manuscript.

2. Clarity and Terminology

The term "Training frequency" appears in Table 1 but lacks explanation. Please clarify its meaning in the table title or description. If it refers to "epoch," consider revising it accordingly.

Ensure all terminology in tables and figures is clearly defined within their titles or content.

Resolve terminology inconsistencies, such as:

"Hollow Space Pyramid Pooling Structure (ASPP)" (page 3) vs. "Atrous Spatial Pyramid Pooling (ASPP)" (page 6).

"Evaluating indicator" (Table 5), "Evaluation index" (Section 3.6 title), and "performance metrics" (Section 3.6) should all be standardized as "Evaluation metrics."

3. Conciseness and Redundancy

Remove unnecessary background explanations. For instance, Figure 2 (U-Net) should be omitted.

Section 3.1 should be condensed into one or two lines.

Section 3.6 should be summarized in two lines and merged into Section 4, such as:

"We use Dice coefficient, Hausdorff distance (HD95), recall, and precision for evaluation."

"HD distance" is incorrect, as it redundantly means "Hausdorff Distance distance." If using HD95 (the standard metric in the field), explicitly specify it.

Reviewer #4: Thank you for your careful revisions. After reviewing the revised manuscript, I am pleased to see that all the concerns raised in the previous round have been adequately addressed. The current version is significantly improved and meets the standards for publication.

**Do you want your identity to be public for this peer review?** For information about this choice, including consent withdrawal, please see our Privacy Policy

Reviewer #3: **Yes: ** Hengrui Zhao

Reviewer #4: No

---

## [Author Response · Author response to Decision Letter 3]

12 May 2025

Dear Reviewer #3:

Thank you for the reviewer’ comments concerning our manuscript entitled “3D-MRI Brain Glioma Intelligent Segmentation Based on Improved 3D U-Net Network” (ID: 1441728). Those comments are all valuable and very helpful for revising and improving our paper, as well as the important guiding significance to our researches. We have studied comments carefully and have made correction which we hope meet with approval.

Responds to the reviewer’s comments:

Reviewer #3:

We feel great thanks for your professional review work on our article. In the revised draft, your suggestions are marked in red. As you are concerned, there are several problems that need to be addressed. According to your nice suggestions, we have made extensive corrections to our previous draft, the detailed corrections are listed below.

1. While the authors revised "Validate the dataset" in one paragraph, it still appears unchanged in Table 1 and Table 3. Please ensure consistency throughout the paper.

Standardize expressions such as "training set," "validation set," and "test set" across all instances."

Data set" should be consistently written as "Dataset."

I recommend using an AI proofreading tool, such as DeepSeek or Qwen, to check and correct grammar errors throughout the manuscript.

Reply: We carefully revised our manuscript. According to your revision suggestions, we made modifications in many places in the manuscript, marked them in red font, and unified "training set," "Dataset," "validation set," and "test set".

2. The term "Training frequency" appears in Table 1 but lacks explanation. Please clarify its meaning in the table title or description. If it refers to "epoch," consider revising it accordingly.

Ensure all terminology in tables and figures is clearly defined within their titles or content.

Resolve terminology inconsistencies, such as:"Hollow Space Pyramid Pooling Structure (ASPP)" (page 3) vs. "Atrous Spatial Pyramid Pooling (ASPP)" (page 6).

"Evaluating indicator" (Table 5), "Evaluation index" (Section 3.6 title), and "performance metrics" (Section 3.6) should all be standardized as "Evaluation metrics."

Reply: We made the modification in red font and changed the "training frequency". And we unified the "Atrous Spatial Pyramid Pooling (ASPP)" and "Evaluation metrics" throughout the text.

3. Remove unnecessary background explanations. For instance, Figure 2 (U-Net) should be omitted.

Section 3.1 should be condensed into one or two lines.

Section 3.6 should be summarized in two lines and merged into Section 4, such as:

"We use Dice coefficient, Hausdorff distance (HD95), recall, and precision for evaluation."

"HD distance" is incorrect, as it redundantly means "Hausdorff Distance distance." If using HD95 (the standard metric in the field), explicitly specify it

Reply: According to your revision suggestions, we made the modifications in red font. We omitted Figure 2 and compressed Section 3.1. We summarize Section3.6 into two lines and merge them into Section 4.We clearly indicated HD95 evaluation metrics.

---

## [Editor Report · Decision Letter 3]

3D-MRI Brain Glioma Intelligent Segmentation Based on Improved 3D U-Net Network

PONE-D-24-42605R3

Dear Dr. Wang,

We’re pleased to inform you that your manuscript has been judged scientifically suitable for publication and will be formally accepted for publication once it meets all outstanding technical requirements.

Kind regards,

Tao Peng

Academic Editor

PLOS ONE

Additional Editor Comments (optional):

Congratulations!
---

## [Editor Report · Acceptance letter]

PONE-D-24-42605R3

PLOS ONE

Dear Dr. Wang,

I'm pleased to inform you that your manuscript has been deemed suitable for publication in PLOS ONE. Congratulations! Your manuscript is now being handed over to our production team.

Kind regards,

on behalf of

Dr. Tao Peng

Academic Editor

PLOS ONE